# BAYESIAN VECTOR OPTIMIZATION WITH GAUSSIAN PROCESSES

## ABSTRACT

Learning problems in which multiple conflicting objectives must be considered simultaneously often arise in various fields, including engineering, drug design, and environmental management. Traditional methods of multi-objective optimization, such as scalarization and identification of the Pareto set under componentwise order, have limitations in incorporating objective preferences and exploring the solution space accordingly. While vector optimization offers improved flexibility and adaptability via specifying partial orders based on ordering cones, current techniques designed for sequential experiments suffer from high sample complexity, which makes them unfit for large-scale learning problems. To address this issue, we propose VOGP, an $(\epsilon, \delta)$-PAC adaptive elimination algorithm that performs vector optimization using Gaussian processes. VOGP allows users to convey objective preferences through ordering cones while performing efficient sampling by exploiting the smoothness of the objective function, resulting in a more effective optimization process that requires fewer evaluations. We first establish provable theoretical guarantees for VOGP, and then derive information gain based and kernel specific sample complexity bounds. VOGP demonstrates strong empirical results on both real-world and synthetic datasets, outperforming previous work in sequential vector optimization and its special case multi-objective optimization. This work highlights the potential of VOGP as a powerful preference-driven method for addressing complex sequential vector optimization problems.

## 1 INTRODUCTION

In diverse fields such as engineering, economics, and computational biology, the problem of identifying the best set of designs across multiple objectives is a recurrent challenge. Often, the evaluation of a particular design is both expensive and noisy, leading to the need for efficient and reliable optimization methods. For instance, in aerospace engineering, the design of a wing structure must balance objectives like aerodynamic drag, wing weight, and fuel weight stored in the wing, each evaluation of which may require costly physical or computational experiments (Obayashi et al., 1997). In the realm of drug discovery, a new compound must be optimized for effectiveness, drug-likeness, and synthesizability, with each trial involving time-consuming lab tests or simulations (Mukaidaisi et al., 2022). A common framework for dealing with such multi-objective optimization problems is to focus on identifying Pareto optimal designs, for which there are no other designs that are better in all objectives. However, this approach can be restrictive, as it only permits a certain set of trade-offs between objectives where a disadvantage on one objective cannot be compensated by advantages on other objectives (i.e., domination relation requires superiority in all dimensions).

A more comprehensive solution to this issue lies in vector optimization, where the partial ordering relations induced by convex cones are used. The core idea is to define a "cone" in the objective space that represents preferred directions of improvement. A solution is considered better than another if the vector difference between their objectives falls within this preference cone. The use of ordering cones in this way gives users a framework that can model a wide range of trade-off preferences (Jahn, 2011).

Let us consider an example. A farmer is planning for the next season and has two key objectives: to maximize crop yield ($Y$) and to minimize water usage ($W$). Water usage is something the farmer wants to minimize, so we can convert it into a maximization problem by considering the negative of

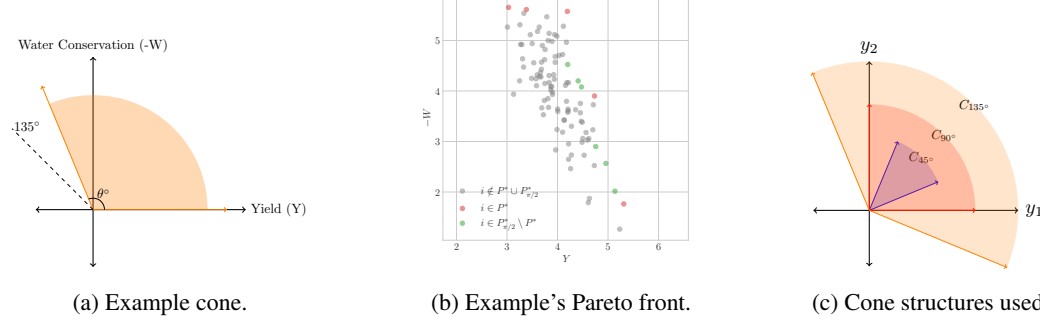

(a) Example cone.      (b) Example's Pareto front.      (c) Cone structures used.

Figure 1: An example cone (1a), its Pareto front (1b) and three different cones used in experiments: $C_{45°}$, $C_{90°}$, and $C_{135°}$ (1c).

water usage $(-W)$. The two objectives become: maximize $Y$ and maximize $-W$. If the farmer is in a region where water resources are scarce, then they might want to prioritize water conservation over crop yield. However, they still cannot disregard yield entirely, as it directly impacts their income. Then, the use of a wide cone that has the following boundaries is appropriate: a boundary that has $0°$ angle with Y axis and another boundary that makes $90°$ to $135°$ degrees with $Y$ axis. The visualization of this cone can be seen in Figure 1a. This indicates the preference for water conservation over crop yield, but not an extreme one. In Figure 1b, it can be seen that some of the designs (shown in green) in the Pareto front of traditional multiobjective order are not in the Pareto front of vector optimization. That is, they are dominated by designs that have better water conservation and slightly worse crop yield. Apart from conveying trade-off preferences of objectives, as discussed in Ararat & Tekin (2023), employing wider or narrower cones may be used as a control mechanism to tune the size of the returned Pareto optimal set. Choosing a narrower cone, as opposed to the positive orthant cone, $\mathbb{R}^M_+$, can potentially yield a more extensive set of Pareto optimal designs. This enriched set includes not only the designs that would be deemed optimal under the standard positive orthant cone, but also encompasses additional designs. These additional designs are superior to arbitrary selections due to their already demonstrated potential in the narrower cone.

The use of ordering cones extends the definition of the Pareto set to objective values that cannot be improved by another one in the sense of this preference cone (Boţ et al., 2009). While there exists previous work addressing Pareto set identification with respect to a preference cone in noisy environments (Ararat & Tekin, 2023), they tend to require vast amounts of samples from the expensive objective function, making them impractical in real-world scenarios. In this paper, we address the problem of identifying the Pareto set with respect to a preference cone $C$ from a discrete set of designs with a minimum number of noisy black-box function queries. We propose *Vector Optimization with Gaussian processes* (VOGP), an $(\epsilon, \delta)$-*probably approximately correct* (PAC) adaptive elimination algorithm that performs vector optimization by utilizing the modeling power of Gaussian processes. In particular, VOGP leverages confidence regions formed by Gaussian processes to perform cone-dependent, probabilistic elimination and Pareto identification steps, which adaptively explore the Pareto front with Bayesian optimization (Garnett, 2023). Similar to PAL and $\epsilon$-PAL (Zuluaga et al., 2013; 2016) for the multi-objective case (see next subsection), VOGP operates in a fully exploratory setting, where the least certain design is queried at each round. When VOGP is considered with the componentwise order, it can be seen as a variant of PAL and $\epsilon$-PAL that can handle dependent objectives. We prove strong theoretical guarantees on the convergence of this algorithm and derive an information gain-based sample complexity bound. Our main contributions can be listed as follows:

- We propose VOGP, an $(\epsilon, \delta)$-PAC adaptive elimination algorithm that performs vector optimization while utilizing Gaussian processes. This is the first work that considers vector optimization within the framework of Bayesian Optimization.

- We theoretically prove that VOGP returns an $(\epsilon, \delta)$-PAC Pareto set with a cone dependent sample complexity bound in terms of the maximum information gain. We also provide kernel specific sample complexity bounds in $\mathcal{O}(\cdot)$ notation.

- We empirically show that VOGP satisfies the theoretical guarantees across different datasets and ordering cones. We also demonstrate that VOGP outperforms existing methods on vector optimization and its special case multi-objective optimization.

**Related work:** There is considerable existing work on Pareto set identification that utilizes Gaussian processes (Lyu et al., 2018; Suzuki et al., 2020; Shu et al., 2020; Mathern et al., 2021; Picheny, 2013; Emmerich et al., 2011; Svenson & Santner, 2016). Some of these works use information theoretic acquisition functions. For instance, PESMO tries to sample designs that optimize the mutual information between the observation and the Pareto set, given the dataset (Hernandez-Lobato et al., 2016). MESMO tries to sample designs that maximize the mutual information between observations and the maximum value of the function (Belakaria et al., 2019). JESMO tries to sample designs that maximize the mutual information between observations and the joint distribution of the optimal points (Tu et al., 2022). Different from entropy search-based methods, some other methods utilize high-probability confidence regions formed by the Gaussian process posterior. For instance, PALS (Barracosa et al., 2022), PAL and $\epsilon$-PAL are confidence region-based adaptive elimination algorithms that aim to categorize the input data points into three groups using the models they have learned: those that are Pareto optimal, those that are not Pareto optimal, and those whose statuses are uncertain (Zuluaga et al., 2013; 2016). In every iteration, these algorithms choose a potential input for evaluation with the aim of reducing the quantity of data points in the uncertain category. Thanks to GP-based modeling, a substantial number of designs can be explored by a single query. Assuming independent objectives, PAL and $\epsilon$-PAL work in the fixed confidence setting and accommodates a variable sampling budget. On the other hand, entropy search-based methods are often investigated in fixed-budget setting without any theoretical guarantee on the accuracy of Pareto set identification. However, all the algorithms mentioned above try to identify the Pareto set according to the componentwise order. They are not able to capture user preferences encoded by ordering cones.

In the context of multi-objective Bayesian optimization, there is a limited amount of work that incorporates user preferences. Abdolshah et al. (2019) propose MOBO-PC, a multi-objective Bayesian optimization algorithm which incorporates user preferences in the form of preference-order constraints. MOBO-PC uses expected Pareto hypervolume improvement weighted by the probability of obeying preference-order constraints as its acquisition function. Yang et al. (2016a) proposes a method that uses truncated expected hypervolume improvement and considers the predictive mean, variance, and the preferred region in the objective space. Some works employ preference learning (Chu & Ghahramani, 2005; Hakanen & Knowles, 2017; Taylor et al., 2021; Ungredda & Branke, 2023; Ignatenko et al., 2021), where the user interacts sequentially with the algorithm to learn the user preference (Astudillo & Frazier, 2019; Lin et al., 2022). Astudillo & Frazier (2019) employ Bayesian preference learning where the user's preferences are modeled as a utility function and they propose two novel acquisition functions that are robust to utility uncertainty. Lin et al. (2022) consider various preference exploration methods while also representing user preferences with a utility function which is modeled with a Gaussian Process. Ahmadianshalchi et al. (2023) propose PAC-MOO, a constrained multi-objective Bayesian optimization algorithm which incorporates preferences in the form of weights that add up to one. PAC-MOO uses the information gained about the optimal constrained Pareto front weighted by preference weights as its acquisition function. Khan et al. (2022) propose a method that learns a utility function offline by using expert knowledge to avoid the repeated and expensive expert involvement.

There are techniques that transform a multi-objective optimization problem into a single-objective problem by assigning weights to the objectives (Ponweiser et al., 2008; Knowles, 2006). The transformed problem can be solved using standard single-objective optimization methods. ParEGO introduced in Knowles (2006) transforms the multi-objective optimization problem into a single-objective one by using Tchebycheff scalarization and solves it by Efficient Global Optimization (EGO) algorithm, which was designed for single-objective expensive optimization problems.

There are existing works that incorporate polyhedral structures to guide the design identification task. As an example, Katz-Samuels & Scott (2018) introduce the concept of feasible arm identification. Their objective is to identify arms whose average rewards fall within a specified polyhedron, using evaluations that are subject to noise. Ararat & Tekin (2023) propose a Naïve Elimination algorithm for Pareto set identification with polyhedral ordering cones. They provide sample complexity bounds on $(\epsilon,\delta)$-PAC Pareto set identification performance of this algorithm. However, their algorithm assumes independent arms and does not perform adaptive elimination. As a result, they only have worst-case

sample complexity bounds for this algorithm. Experiments show that identification requires a large sampling budget which renders Naïve Elimination impractical in real-world problems of interest. Table 1 compares our work with the most closely related prior works in terms of key differences.

Table 1: Comparison with Related Works

| Works | Cone-based preferences | Sample complexity bounds | $(\epsilon,\delta)$-PAC | Utilizes GPs |
|---|---|---|---|---|
| This work | Yes | Yes | Yes | Yes |
| Ararat & Tekin (2023) | Yes | Yes | Yes | No |
| Zuluaga et al. (2016) | No | Yes | Yes | Yes |
| Belakaria et al. (2019) | No | No | No | Yes |
| Hernandez-Lobato et al. (2016) | No | No | No | Yes |
| Tu et al. (2022) | No | No | No | Yes |

## 2 BACKGROUND AND PROBLEM DEFINITION

We consider the problem of sequentially optimizing a vector-valued function $\boldsymbol{f} : \mathcal{X} \to \mathbb{R}^M$ over a finite set of designs $\mathcal{X} \subset \mathbb{R}^D$ with respect to a polyhedral ordering cone $C \subset \mathbb{R}^M$. In each iteration $t$, a design point $x_t$ is selected, and a noisy observation is recorded as $\boldsymbol{y}_t = \boldsymbol{f}(x_t) + \boldsymbol{\nu}_t$. Here, the noise component $\boldsymbol{\nu}_t$ has the multivariate Gaussian distribution $\mathcal{N}\left(\boldsymbol{0}, \sigma^2 \boldsymbol{I}_{M \times M}\right)$, where $\sigma^2$ denotes the variance of the noise. To define the optimality of designs, we use the partial order induced by $C$.

**Partial order induced by a cone** We assume that the partial order among designs is induced by a known polyhedral ordering cone $C$ whose interior is nonempty. Such a cone can be written as $C = \left\{\boldsymbol{x} \in \mathbb{R}^M \mid \boldsymbol{W}\boldsymbol{x} \geq 0\right\}$, where $\boldsymbol{W}$ is a $N \times M$ matrix. $N$ is the number of halfspaces that define $C$. Rows $\boldsymbol{w}_1^\top, \ldots, \boldsymbol{w}_N^\top$ are the unit normal vectors of these halfspaces with respect to the Euclidean norm $\|\cdot\|_2$. For $r \geq 0$, we write $\mathbb{B}(r) := \left\{\boldsymbol{y} \in \mathbb{R}^M \mid \|\boldsymbol{y}\|_2 \leq r\right\}$. We say that $\boldsymbol{y} \in \mathbb{R}^M$ weakly dominates $\boldsymbol{y}' \in \mathbb{R}^M$ with respect to $C$ if their difference lies in $C$: $\boldsymbol{y}' \preccurlyeq_C \boldsymbol{y} \Leftrightarrow \boldsymbol{y} - \boldsymbol{y}' \in C$. By the structure of $C$, an equivalent expression is: $\boldsymbol{y}' \preccurlyeq_C \boldsymbol{y} \Leftrightarrow \boldsymbol{w}_n^\top (\boldsymbol{y} - \boldsymbol{y}') \geq 0 \; \forall n \in [N] := \{1, \ldots, N\}$. The Pareto set with respect to a cone $C$ is the set of designs that are not weakly dominated by another design with respect to $C$: $P_C^* = \{x \in \mathcal{X} \mid \nexists x' \in \mathcal{X} \setminus \{x\} : \boldsymbol{f}(x) \preccurlyeq_C \boldsymbol{f}(x')\}$.

$(\epsilon, \delta)$**-PAC Pareto set** We use the cone-dependent, direction-free suboptimality gaps defined in Ararat & Tekin (2023). The gap between designs $x, x' \in \mathcal{X}$ is given as $m(x, x') := \inf \{s \geq 0 \mid \exists \boldsymbol{u} \in \mathbb{B}(1) \cap C : \boldsymbol{f}(x) + s\boldsymbol{u} \notin \boldsymbol{f}(x') - \text{int}(C)\}$, where $\text{int}(C)$ denotes the interior of $C$ and $-$ operator over sets is the Minkowski difference (vectors being treated as singletons). The suboptimality gap of design $x$ is defined as $\Delta_x^* := \max_{x' \in P^*} m(x, x')$.

**Definition 1.** *Let $\epsilon > 0$, $\delta \in (0, 1)$. A random set $P \subseteq \mathcal{X}$ is called an $(\epsilon, \delta)$-PAC Pareto set if the subsequent conditions are satisfied at least with probability $1 - \delta$:*
*(i) $\bigcup_{x \in P} (\boldsymbol{f}(x) + \mathbb{B}(\epsilon) \cap C - C) \supseteq \bigcup_{x \in P^*} (\boldsymbol{f}(x) - C)$; (ii) $\forall x \in P \setminus P^* : \Delta_x^* \leq 2\epsilon$.*

**Remark 1.** *Condition $(i)$ of Definition 1 is equivalent to the following: For every $x^* \in P^*$, there exist $x \in P$ and $\boldsymbol{u} \in \mathbb{B}(\epsilon) \cap C$ such that $\boldsymbol{f}(x^*) \preccurlyeq_C \boldsymbol{f}(x) + \boldsymbol{u}$. Even though certain designs in P may not be optimal, condition $(ii)$ of Definition 1 limits the extent of the inadequacies in these designs. Consequently, it ensures the overall quality of all the designs produced.*

**Goal** Our aim is to design an algorithm that returns an $(\epsilon, \delta)$-PAC Pareto set $\hat{\mathcal{P}} \subseteq \mathcal{X}$ with as little sampling from expensive objective function $\boldsymbol{f}$ as possible.

$M$**-output Gaussian process** We model the objective function $\boldsymbol{f}$ as a realization of an $M$-output GP with zero mean and a positive definite covariance function $\boldsymbol{k}$ with bounded variance: $k^{jj}(x, x) \leq 1$ for every $x \in \mathcal{X}$ and $j \in [M]$. Let $\tilde{x}_i$ be the $i^\text{th}$ design observed and $\boldsymbol{y}_i$ the corresponding observation. The posterior distribution of $\boldsymbol{f}$ conditioned on the first $t$ observations is that of an $M$-output GP with mean ($\boldsymbol{\mu}_t$) and covariance ($\boldsymbol{k}_t$) functions given below, where $\boldsymbol{k}_t(x) = [\boldsymbol{k}(x, x_1), \ldots, \boldsymbol{k}(x, x_t)] \in \mathbb{R}^{M \times Mt}$, $\boldsymbol{y}_{[t]} = \left[\boldsymbol{y}_1^\top, \ldots, \boldsymbol{y}_t^\top\right]^\top$, $\boldsymbol{K}_t = (\boldsymbol{k}(x_i, x_j))_{i,j \in [t]}$, and $\boldsymbol{I}_{Mt}$ is the $Mt \times Mt$ identity matrix:

$$\boldsymbol{\mu}_t(x) = \boldsymbol{k}_t(x) \left(\boldsymbol{K}_t + \sigma^2 \boldsymbol{I}_{Mt}\right)^{-1} \boldsymbol{y}_{[t]}^\top,$$

$$\boldsymbol{k}_t(x, x') = \boldsymbol{k}(x, x') - \boldsymbol{k}_t(x) \left(\boldsymbol{K}_t + \sigma^2 \boldsymbol{I}_{Mt}\right)^{-1} \boldsymbol{k}_t(x')^\top.$$

**Definition 2.** *The maximum information gain at round $t$ is defined as $\gamma_t := \max_{A \subseteq \mathcal{X}:|A|=t} \mathrm{I}(\boldsymbol{y}_A; \boldsymbol{f}_A)$, where $\boldsymbol{y}_A$ is the collection of observations corresponding to the designs in $A$, $\boldsymbol{f}_A$ is the collection of the corresponding function values, and $\mathrm{I}(\boldsymbol{y}_A; \boldsymbol{f}_A)$ is the mutual information between the two.*

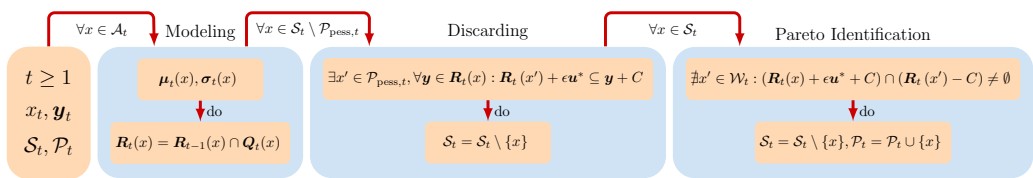

Figure 2: Illustration of VOGP algorithm's phases.

## 3 THE ALGORITHM: VOGP

We propose *Vector Optimization with Gaussian processes* (VOGP), an $(\epsilon, \delta)$-PAC adaptive elimination algorithm that performs vector optimization by utilizing the modeling power of GPs. VOGP takes as input the design set $\mathcal{X}$, cone $C$, accuracy parameter $\epsilon$, and confidence parameter $\delta$. VOGP adaptively classifies designs as suboptimal, Pareto optimal, and uncertain. Initially, all designs are put in the set of uncertain designs, denoted by $\mathcal{S}_t$. Designs that are classified as suboptimal are discarded and never considered again in comparisons. Designs that are classified as Pareto optimal are moved to the predicted Pareto set $\hat{\mathcal{P}}_t$. VOGP terminates and returns $\hat{\mathcal{P}} = \hat{\mathcal{P}}_t$ when the set of uncertain designs becomes empty. Decision-making in each round is divided into four phases named *modeling*, *discarding*, *Pareto identification*, *evaluating*. An illustration of these phases is given in Figure 2. Below, we discuss each phase in detail.

---

**Algorithm 1** VOGP Algorithm

---

1: **Input:** Design set $\mathcal{X}$, accuracy parameter $\epsilon \geq 0$, GP prior $\boldsymbol{\mu}_0, \boldsymbol{\sigma}_0$, polyhedral ordering cone $C$
2: **Initialize:** $\mathcal{P}_1 = \emptyset, \mathcal{S}_1 = \mathcal{X}, \boldsymbol{R}_0(x) = \mathbb{R}^m$ for each $x \in \mathcal{S}_1, t = 1$
3: **Compute:** Accuracy vector $\boldsymbol{u}^* \in C$
4: **while** $\mathcal{S}_t \neq \emptyset$ **do**
5:     MODELING($\mathcal{S}_t, \mathcal{P}_t$);
6:     DISCARDING($\mathcal{S}_t$);
7:     PARETOIDENTIFICATION($\mathcal{S}_t, \mathcal{P}_t$);
8:     EVALUATING($\mathcal{S}_t, \mathcal{W}_t$);
9:     $\mathcal{P}_{t+1} = \mathcal{P}_t, \mathcal{S}_{t+1} = \mathcal{S}_t$;
10:     $t = t + 1$;
11: **end while**
12: **return** Predicted Pareto set $\hat{\mathcal{P}} = \mathcal{P}_t$

---

**Modeling** VOGP uses the GP posterior means and variances of designs to define confidence regions in the form of $M$-dimensional hyperrectangles which are scaled by a function of $\delta$. The probability that these hyperrectangles include the true objective values of the designs is at least $1 - \delta$, where $\delta$ is the confidence parameter of the algorithm. Using these confidence hyperrectangles allows for the formulation of probabilistic success criteria and the quantification of uncertainty with designs. The hyperrectangles are defined by (1) below, where $\beta_t$ is a scalar chosen to ensure that the hyperrectangles include the true objective values at least with probability $1 - \delta$. We take $\beta_t = 2 \ln(M\pi^2 |\mathcal{X}| t^2 / (3\delta))$.

$$\boldsymbol{Q}_t(x) = \left\{ \boldsymbol{y} \in \mathbb{R}^M \mid \mu_t^j(x) - \beta_t^{1/2} \sigma_t^j(x) \leq y^j \leq \mu_t^j(x) + \beta_t^{1/2} \sigma_t^j(x), \forall j \in \{1, 2, \ldots, M\} \right\} . \quad (1)$$

At each round, the calculated hyperrectangles are intersected cumulatively to obtain the cumulative confidence intervals for the designs. Once the algorithm has modeled the designs using confidence hyperrectangles, it proceeds to the discarding phase.

---

**Algorithm 2** Modeling Subroutine

---

1: $\mathcal{A}_t = \mathcal{S}_t \cup \mathcal{P}_t$;
2: **for** $x \in \mathcal{A}_t$ **do**
3:     Obtain GP posterior for $x$: $\boldsymbol{\mu}_t(x)$ and $\boldsymbol{\sigma}_t(x)$;
4:     Construct the confidence hyperrectangle $\boldsymbol{Q}_t(x)$;
5:     $\boldsymbol{R}_t(x) = \boldsymbol{R}_{t-1}(x) \cap \boldsymbol{Q}_t(x)$;
6: **end for**

---

**Discarding** VOGP discards undecided designs that are dominated with respect to $C$ with high probability. In order to speed up the discarding phase, the designs that are to be discarded with low probability are identified and are used to form the pessimistic Pareto set $\mathcal{P}_{\text{pess},t}^{(C)}(\mathcal{S}_t)$ as defined below.

**Definition 3** (Pessimistic Pareto Set). *Let $t \geq 1$ and let $D \subseteq \mathcal{A}_t$ be a set of nodes. The* pessimistic Pareto set *of $D$ with respect to $C$ at round $t$, denoted by $\mathcal{P}_{\text{pess},t}^{(C)}(D)$, is the set of all nodes $x \in D$ for which there is no other node $x' \in D \setminus \{x\}$ such that $\boldsymbol{R}_t(x') \subseteq \boldsymbol{R}_t(x) + C$.*

VOGP computes pessimistic Pareto set of $\mathcal{S}_t$ by checking for each design in $\mathcal{S}_t$ whether there is another design that prevents it from being in the pessimistic Pareto set. To see if design $x$ prevents design $x'$ from being in the pessimistic Pareto set, by solving a convex optimization problem, VOGP checks if every vertex of $\boldsymbol{R}_t(x')$ can be expressed as the sum of a vector in $C$ and a vector in $\boldsymbol{R}_t(x)$.

VOGP refines the undecided set by utilizing the pessimistic Pareto set. It eliminates designs based on the following criteria: for a given design $x$, if all values inside its confidence hyperrectangle, when added to an "accuracy vector", are dominated with respect to $C$ by all values in the confidence hyperrectangle of another design $x'$, then the dominated design $x$ is discarded. The dominating design $x'$ should be a member of the pessimistic Pareto set.

The accuracy vector is of the form $\epsilon \boldsymbol{u}^*$, where the accuracy parameter $\epsilon$ gives its norm and $\boldsymbol{u}^*$ is a unit direction vector, called the accuracy direction. The latter is selected strategically to maximize the robustness of the ordering relation induced by $C$. More precisely, the accuracy direction dominates the origin, even with a small perturbation in an arbitrary direction $\boldsymbol{u} \in \mathbb{S}^{M-1}$, where $\mathbb{S}^{M-1}$ denotes the unit sphere in $\mathbb{R}^M$ (see Lemma 1 below). For this purpose, $\boldsymbol{u}^*$ is defined as follows:

**Definition 4.** *Let $A(1) := \bigcap_{\boldsymbol{u} \in \mathbb{S}^{m-1}} (\boldsymbol{u} + C)$ and $d(1) := \inf\{\|\boldsymbol{z}\|_2 \mid \boldsymbol{z} \in A(1)\}$. Then, there exists a unique vector $\boldsymbol{z}^* \in A(1)$ such that $d(1) = \|\boldsymbol{z}\|_2$ and we define $\boldsymbol{u}^* := \frac{\boldsymbol{z}^*}{d(1)}$.*

**Lemma 1.** *For every $\boldsymbol{y}, \boldsymbol{z} \in \mathbb{R}^M$ and $\tilde{\boldsymbol{p}} \in \mathbb{B}(\frac{\epsilon}{d(1)})$, having $\boldsymbol{y} + \tilde{\boldsymbol{p}} \preccurlyeq_C \boldsymbol{z}$ implies $\boldsymbol{y} \preccurlyeq_C \boldsymbol{z} + \epsilon \boldsymbol{u}^*$.*

---

**Algorithm 3** Discarding Subroutine

---

1: **Compute:** $\mathcal{P}_{\text{pess},t} = \mathcal{P}_{\text{pess},t}^{(C)}(\mathcal{S}_t)$
2: **for** $x \in \mathcal{S}_t \setminus \mathcal{P}_{\text{pess},t}$ **do**
3:     **if** $\exists x' \in \mathcal{P}_{\text{pess},t} \ \forall \boldsymbol{y} \in \boldsymbol{R}_t(x) : \boldsymbol{R}_t(x') + \epsilon \boldsymbol{u}^* \subseteq \boldsymbol{y} + C$ **then**
4:         $\mathcal{S}_t = \mathcal{S}_t \setminus \{x\}$;
5:     **end if**
6: **end for**

---

**Pareto identification** VOGP aims to identify the designs that are not dominated by any other design with high probability with respect to the ordering cone $C$. It does so by pinpointing designs that, after adding the accuracy vector to the values in their confidence hyperrectangles, remain non-dominated with respect to $C$ when compared to the values in the confidence hyperrectangle of any other design.

The identified designs are moved from the set of undecided designs to the predicted Pareto set. It is important to note that, once a design becomes a member of the predicted Pareto set, it remains a permanent member of that set.

---

**Algorithm 4** Pareto Identification Subroutine

---
1: $\mathcal{W}_t = \mathcal{S}_t \cup \mathcal{P}_t$;
2: **for** $x \in \mathcal{S}_t$ **do**
3:     **if** $\nexists x' \in \mathcal{W}_t : (\boldsymbol{R}_t(x) + \epsilon \boldsymbol{u}^* + C) \cap \boldsymbol{R}_t(x') \neq \emptyset$ **then**
4:         $\mathcal{S}_t = \mathcal{S}_t \setminus \{x\}, \mathcal{P}_t = \mathcal{P}_t \cup \{x\}$;
5:     **end if**
6: **end for**

---

**Evaluating** VOGP selects the design whose confidence hyperrectangle has the widest diagonal. The diagonal of the hyperrectangle $\boldsymbol{R}_t(x)$ is given by $\omega_t(x) = \max_{\boldsymbol{y}, \boldsymbol{y}' \in \boldsymbol{R}_t(x)} \|\boldsymbol{y} - \boldsymbol{y}'\|_2^2$. The motivation behind this step is to acquire as much information about the objective space as possible, so that the distinction between Pareto and non-Pareto designs can be made fast with high probability.

---

**Algorithm 5** Evaluating Subroutine

---
1: **if** $\mathcal{S}_t \neq \emptyset$ **then**
2:     Select design $x_t \in \arg\max_{x \in \mathcal{W}_t} \omega_t(x)$ (break ties randomly);
3:     Observe $\boldsymbol{y}_t = \boldsymbol{f}(x_t) + \boldsymbol{\nu}_t$;
4: **end if**

---

## 4   Theoretical Analysis

In this section, we provide theoretical guarantees for VOGP and sketch their derivation. Existing adaptive elimination methods on multi-objective optimization compare designs in componentwise order. Our proof techniques diverge from the this line of prior work by elaborately using properties of the ordering cone.

**Theorem 1.** *Let $\eta := \sigma^{-2} / \ln\left(1 + \sigma^{-2}\right)$. When VOGP is run, the following holds at least with probability $1 - \delta$: An $(\epsilon, \delta)$-Pareto set can be identified with no more than $T$ function evaluations, where*

$$T := \min \left\{ t \in \mathbb{N} : \sqrt{\frac{8\beta_t \sigma^2 \eta M \gamma_t}{t}} \leq \frac{\epsilon}{2d(1)} \right\} . \tag{2}$$

Here, we sketch the idea of the proof. The full proof can be found in supplemental document. First, we define an event $E$ under which the confidence hyperrectangles of designs include their true objective values. We prove that $\mathbb{P}(E) \geq 1 - \delta$. We introduce a new cone-dependent accuracy vector $\epsilon \boldsymbol{u}^*$ as a means of performing comparisons under uncertainty. Next, we derive some useful properties of $d(1)$ and $A(1)$ (see Definition 4). Then, by leveraging cone domination properties together with the subroutines of the algorithm, we establish that the set $\hat{\mathcal{P}}$ returned by VOGP is an $(\epsilon, \delta)$-PAC Pareto set (see Definition 1). We determine the stopping criterion for VOGP in a unique way, by utilizing the uncertainty in designs based on the properties of $\epsilon \boldsymbol{u}^*$ (see Lemma 1). We use the stated stopping criterion to derive an upper bound on the sample complexity of VOGP.

**Theorem 2.** *Let the GP kernel $\boldsymbol{k}$ have the multiplicative form $\boldsymbol{k}(x, x') = [\tilde{k}(x, x') k^*(p, q)]_{p,q \in [M]}$, where $\tilde{k} \colon \mathcal{X} \times \mathcal{X} \to \mathbb{R}$ is a kernel for the design space and $k^* \colon [M] \times [M] \to \mathbb{R}$ is a kernel for the objective space. Assume that $\tilde{k}$ is a squared exponential kernel or a Matérn kernel. Let $\Lambda := 4\alpha_2 d^2(1)\left(2 + \ln(\alpha_1)\right), \alpha_1 := M\pi^2 |\mathcal{X}| / (3\delta), \alpha_2 := 16\sigma^2 \eta M^2$. Then, the sample complexity of VOGP is given by*

$$T = \mathcal{O} \left( e^{\left(\tau \sqrt{2\left(\ln(\tau) - \frac{\ln(\omega)}{\tau} - 1\right)}\right)} \cdot \frac{\tau^\tau}{\omega} \right) . \tag{3}$$

Here, $\tau = D + 2$ and $\omega = \epsilon^2 / \Lambda$ when $\tilde{k}$ is a squared exponential kernel; $\tau = (D + 2)(2\nu + D(D + 1))/2\nu$ and $\omega = (\epsilon^2 / \Lambda)^{\frac{2\nu + D}{2\nu}}$ when $\tilde{k}$ is a Matérn kernel with smoothness parameter $\nu$.

To prove Theorem 2, we use some bounds on the mutual information of the observations and the true values for an $M$-output GP to bound the $\gamma_t$ term in (2). Then, we find an upper bound on the sample complexity which is expressed in terms of the Lambert $W$ function. Finally, we use some tight bounds for $W$ in Chatzigeorgiou (2013a) to bound the sample complexity of VOGP.

**Remark 2.** *Theorem 1 provides a bound on sample complexity that is applicable in a very general context where the kernel $\boldsymbol{k}$ can be any bounded positive definite kernel. However, since it has the form of an $\mathrm{argmin}$ over an inequality, it may be hard to interpret. Theorem 2 provides a more explicit and practical formulation of sample complexity that shows how it scales in relation to VOGP's parameters.*

## 5 EXPERIMENTS

We evaluate the performance of VOGP on two synthetic and two real-world datasets. We investigate its adherence to the proven theoretical guarantees (success rates SR1, SR2, function evaluation count (SC), and Pareto statistics (namely, Pareto accuracy (PA), recall (PR), and precision (PP) values in classifying the designs as Pareto or not.) We will investigate the performance of VOGP under $C_{45°}$, $C_{90°}$, and $C_{135°}$ in comparison to the state of the art method for vector optimization: Naïve Elimination (NE). There are many established multi-objective optimization (MOO) methods mentioned in the related works section. However, adapting these methods to general cone structures is non-trivial. A comparison with these methods is still possible since VOGP can be used for MOO purposes by choosing $C = C_{90°}$. Therefore, we provide an extensive comparison of VOGP with other state-of-the-art MOO methods in the context of $C_{90°}$ in the supplemental document (Section A.1). These methods are MESMO, JESMO, PESMO and EHVI (Daulton et al., 2020). The reported results are obtained by repeating all experiments 10 times and averaging the scores. The results of the experiments can be seen in Table 2.

- **SNW dataset** This dataset was introduced by Zuluaga et al. (2012). The objective here is to optimize the area and throughput for the synthesis of a field-programmable gate array (FPGA) platform ($M = 2, D = 3$). The dataset comprises 206 distinct hardware design implementations of a sorting network.

- **BC (Branin Currin) dataset** For this dataset, we utilize two commonly used benchmark functions, namely Branin and Currin, each serving as an output dimension. The input space for this dataset is two-dimensional ($M = 2, D = 2$).

- **OKA dataset** This dataset involves the evaluation of two functions as defined in Okabe et al. (2004) ($M = 2, D = 3$).

- **SnAr (chemical reaction) dataset** This benchmark examines the nucleophilic aromatic substitution reaction (SnAr) involving 2,4-difluoronitrobenzene and pyrrolidine in ethanol, which results in a primary product and two additional side-products (Hone et al., 2017). The objectives are the space-time yield and the environmental impact. The dataset comprises 950 designs ($M = 2, D = 4$).

- **MAR (marine) dataset** This dataset focuses on optimizing bulk carrier architectures while obeying specific restrictions required for vessels navigating the Panama Canal (Parsons & Scott, 2004). The objective includes maximizing annual cargo, while minimizing transportation cost and ship weight. The mentioned constraints are also transformed into objectives. The dataset comprises of 1000 designs ($M = 4, D = 6$).

**Definition 5.** *Given an angle $\theta°$, the cone $C_{\theta°} \subseteq \mathbb{R}^2$ is defined as the convex cone whose two boundary rays make $\frac{\pm\theta}{2}$ degrees with the identity line.*

**Definition 6.** *Given a finite set of designs $\mathcal{X}$, set of Pareto designs $P_\theta^*$ and a set of predicted Pareto designs $P_\theta$ with respect to the cone $C_{\theta°}$, Pareto accuracy (PA), Pareto recall (PR), and Pareto precision (PP), rate of satisfying success conditions $(i)$ (SR1) and $(ii)$ (SR2) are defined as*

$$\mathrm{PA} := \frac{|P_\theta^* \cap P_\theta| + |(\mathcal{X} \backslash P_\theta^*) \cap (\mathcal{X} \backslash P_\theta)|}{|\mathcal{X}|} \times 100 \,, \quad \mathrm{PR} := \frac{|P_\theta^* \cap P_\theta|}{|P_\theta^*|} \times 100 \,, \quad \mathrm{PP} := \frac{|P_\theta^* \cap P_\theta|}{|P_\theta|} \times 100$$

$$\mathrm{SR1} := \frac{\left|\left\{x \in P_\theta^* : \boldsymbol{f}(x^*) \in \bigcup_{x \in P_\theta} (\boldsymbol{f}(x) + \mathbb{B}(\epsilon) \cap C - C)\right\}\right|}{|P_\theta^*|} \,, \quad \mathrm{SR2} := \frac{|\{x \in P_\theta : \Delta_x^* \le 2\epsilon\}|}{|P_\theta|} \,.$$

**Experimental Setup** Prior to the experiments, we normalize the datasets for $\epsilon$ parameter to be applicable. Since we normalize the datasets, we apply a noise standart deviation of $0.1$. We fix $\epsilon = 0.1$ and $\delta = 0.05$. Similar to the other works with the same confidence hyperrectangle definition, we scale down $\beta_t$ by a factor of 20 (Zuluaga et al. (2016)). Initially, for each dataset, we learn the kernel hyperparameters by training on the entire dataset. This is done to ensure that the algorithm

capture the smoothness of the dataset correctly. Data used in this phase is not used to update the GP posterior. All algorithms start the learning phase with an empty set of observations. In this experiment, for OKA dataset, Matérn kernel was used. For SNW, BC, SnAr datasets, RBF kernel was used. NE is a fixed budget algorithm and it takes at least one sample from each design. Prior work derived a theoretically sufficient sampling budget for NE under which an $(\epsilon, \delta)$-PAC Pareto set is guaranteed to be returned; however, this budget is too large to be useful in practice. To make results comparable, NE was given a sample-per-design budget that makes total number of evaluations as close as possible $(\lceil T/|\mathcal{X}|\rceil)$ .

**Discussion** As seen in Table 2, despite the disparity in sampling budgets, VOGP performs very similarly but moderately below NE in terms of Pareto accuracy (PA), Pareto precision (PP), and Pareto recall (PR). VOGP performs marginally below NE in terms of satisfying the $(\epsilon,\delta)$-PAC conditions. These can be attributed to the disparity between the sample counts between NE and VOGP. Since NE takes at least one sample from each design, in cases where VOGP samples less designs than $|\mathcal{X}|$, NE gets to sample more designs than VOGP. As can be confirmed from Table 2, this is a lot more than VOGP's total sample count for most cases; specifically, for all cases except when SNW dataset is used with $C_{45°}$. As an example, for SnAr dataset with $C_{135°}$, VOGP had a total budget of around 33 whereas NE had 950. As can be seen in Table 2, despite NE's advantage of sample counts, VOGP was capable of achieving superior $(\epsilon,\delta)$-PAC success rates for multiple settings and comparable Pareto statistics while requiring significantly fewer samples in almost all of the cases. This is an expected result because VOGP utilizes an $M$-output GP to obtain information from close observations while also taking correlations of output dimensions into account whereas NE uses empirical means only.

Table 2: Performance comparison between VOGP (Our method) and Naive Elimination under $C_{45°}$, $C_{90°}$, and $C_{135°}$. SR1 and SR2: The success rate of satisfying $(\epsilon,\delta)$-PAC conditions $(i)$ and $(ii)$ respectively. PA: Pareto accuracy rate. PR: Pareto recall rate. PP: Pareto precision rate. SC: The number of evaluations.

| | | VOGP | | | | | | Naive Elimination | | | | | |
|---|---|---|---|---|---|---|---|---|---|---|---|---|---|
| $C$ | D | SR1 ↑ | SR2 ↑ | PA ↑ | PR ↑ | PP ↑ | SC | SR1 ↑ | SR2 ↑ | PA ↑ | PR ↑ | PP ↑ | SC |
| $C_{45°}$ | SNW | 95.96 ±2.78 | 98.65 ±1.46 | 85.53 ±1.35 | **76.54 ±4.93** | 69.49 ±3.25 | 777.8 ±220.65 | **96.92 ± 3.87** | **100.00 ±0.00** | **88.40±2.42** | 72.31 ±6.50 | **79.95±5.21** | 3×250 |
| $C_{45°}$ | BC | **95.17 ±7.43** | 99.37 ±1.27 | **95.36 ±1.59** | **86.90 ±8.13** | **76.56 ±6.35** | 446.40 ±131.14 | 91.03±3.52 | **99.63±1.11** | 92.60±1.26 | 67.93±8.45 | 68.25±5.16 | 2×250 |
| $C_{45°}$ | OKA | 88.00 ±7.38 | 92.84 ±5.41 | 95.28 ±0.99 | 76.40 ±4.54 | 76.87 ±6.89 | 650.3 ±153.36 | **98.00±2.00** | **100.00±0.00** | **96.56±1.11** | **82.40±7.42** | **83.77±7.61** | 3×250 |
| $C_{45°}$ | SnAr | 78.75 ±14.03 | **98.00 ±6.00** | 98.46 ±0.42 | **55.62 ±16.64** | 53.63 ±12.25 | 517.0 ±148.79 | **88.12 ± 5.90** | 97.86 ± 3.27 | **98.56±0.22** | 55.00±6.12 | **58.52±8.07** | 1×950 |
| $C_{90°}$ | SNW | **94.62 ±4.28** | 97.5 ±2.22 | 89.85 ±2.14 | **63.85 ±10.35** | 59.61 ±8.75 | 112.70 ±28.49 | 92.69±7.78 | **99.23±1.54** | **90.00±1.19** | 56.54±7.10 | **61.23±4.96** | 1×250 |
| $C_{90°}$ | BC | **98.33 ±5.00** | **100.0 ±0.00** | **99.20 ±0.59** | 80.00 ±20.82 | **88.33 ±14.53** | 33.60 ±2.73 | **98.33±5.00** | **100.00 ± 0.00** | 98.32±0.61 | 68.33±17.40 | 67.03±15.97 | 1×250 |
| $C_{90°}$ | OKA | 87.14 ±4.29 | **99.09 ±2.73** | 96.24 ±0.57 | 57.14 ±6.39 | 38.91 ±5.84 | 110.60 ±16.38 | **90.00 ±11.16** | 98.33 ±5.00 | **97.96±0.66** | **60.00±16.66** | **68.93±16.40** | 1×250 |
| $C_{90°}$ | SnAr | 67.5 ±40.39 | 95.00 ±10.00 | **99.46 ±0.14** | 40.00 ±25.50 | 34.42 ±21.34 | 39.80 ±20.26 | **95.00 ± 15.00** | **98.57 ± 4.29** | 99.43±0.18 | **55.00±10.00** | **40.19±11.51** | 1×950 |
| $\mathbb{R}^4_+$ | MAR | **100.00 ±0.00** | **99.67 ±0.81** | **97.70 ±0.31** | **79.32 ±2.96** | **66.84 ±4.29** | 716.43 ±173.89 | 98.42 ± 1.29 | 99.00 ± 1.22 | 96.88±0.51 | 71.58±5.98 | 57.53±6.21 | 1×1000 |
| $C_{135°}$ | SNW | **92.00 ±9.80** | 94.39 ±6.41 | 96.12 ±1.76 | **72.00 ±12.49** | 60.07 ±15.58 | 72.80 ±11.91 | 89.00 ±11.36 | **100.00 ± 0.00** | **96.41±1.13** | 63.00±13.45 | **64.19±13.28** | 1×250 |
| $C_{135°}$ | BC | **100.00 ±0.00** | **100.00 ±0.00** | **99.88 ±0.18** | **85.00 ±22.91** | **90.00 ±22.91** | 17.50 ±2.77 | **100.00± 0.00** | **100.00± 0.00** | 99.76±0.20 | **85.00±22.91** | 90.00±15.28 | 1×250 |
| $C_{135°}$ | OKA | 65.00 ±22.91 | 85.00 ±22.91 | 99.60 ±0.31 | 65.00 ±22.91 | 85.00 ±22.91 | 44.60 ±5.22 | **95.00±15.00** | **100.00 ± 0.00** | **99.96±0.12** | **95.00±15.00** | **100.00 ± 0.00** | 1×250 |
| $C_{135°}$ | SnAr | 73.33 ±13.33 | **100.0 ±0.0** | 99.64 ±0.10 | 26.67 ±20.00 | 38.33 ±30.78 | 32.90 ±9.41 | **83.33 ± 30.73** | **100.00 ± 0.00** | **99.75±0.15** | **43.33±26.03** | **65.00±39.76** | 1×950 |

# 6 Conclusion, Limitations and Future Research

We consider black-box vector optimization with noisy evaluations where the objective function is expensive to evaluate. We propose a sample-efficient adaptive elimination algorithm, VOGP. We prove that VOGP returns an $(\epsilon, \delta)$-PAC Pareto set and derive information gain-based and kernel-specific sample complexity bounds.

In our experiments, we observe that VOGP outperforms NE in terms of sampling budget needed to return an $(\epsilon, \delta)$-PAC Pareto set. Furthermore, VOGP requires significantly fewer function evaluations to obtain Pareto statistics similar to those of NE. In the experiments given in the supplemental document, we observe that VOGP is the most consistently high-performing method among other MOO methods across the datasets and metrics presented. In real-world datasets (SnAr, SNW), the lead of VOGP over MOO methods (JESMO, MESMO, PESMO, EHVI) is amplified. In conclusion, VOGP surpasses the state-of-the-art methods of vector optimization and the specific case of multi-objective optimization in terms of the needed function evaluation count and returned Pareto set quality.

Our work opens up many research directions in which vector optimization with Gaussian Processes can be improved. One of those directions is to extend VOGP to continuous design setting, which would require a more nuanced theoretical analysis and incorporating techniques like adaptive discretization. Another direction worth exploring is extending VOGP to the case of missing rewards where observations have incomplete objective values. This could broaden its applicability to real-world problems where data might be partial or missing.

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

SUPPLEMENTAL DOCUMENT

**Overview** In Section A, we present additional experiments by providing the comparison of VOGP with other MOO methods. In Section B, we provide the comparison of VOGP and random search. In Section C, we present an ablation study where the absence of known kernel hyperparameters assumption is investigated. In Section D, we provide an extension of our related works that encapsulates related works on evolutionary algorithms. In Section E, we provide the derivations of Theorems 1, 2.

# A    COMPARISON OF VOGP WITH OTHER MOO METHODS

## A.1    COMPARISON OF VOGP WITH OTHER MOO METHODS UNDER $C = C_{90°}$

Here, we undertake a comparative study of the VOGP algorithm and other MOO methods within the framework of a $90°$ ordering cone. These methods are MESMO, JESMO, PESMO and EHVI. Our objective is to compare the performance of algorithms in terms of their sample complexity and the quality of the resulting Pareto front.

Table 3: Comparison of VOGP with other MOO methods under $C_{90°}$. SR1 and SR2: The success rate of satisfying $(\epsilon,\delta)$-PAC conditions $(i)$ and $(ii)$ respectively. HV: Hypervolume. PA: Pareto accuracy rate. PR: Pareto recall rate. PP: Pareto precision rate. SC: The number of evaluations.

| Method | Dataset | SR1 ↑ | SR2 ↑ | HV ↑ | PA ↑ | PR ↑ | PP ↑ | SC↓ |
|---|---|---|---|---|---|---|---|---|
| VOGP (Ours) | SNW | **94.62** ±4.28 | **97.5** ±2.22 | **45.89** ±0.09 | **89.85** ±2.14 | **63.85** ±10.35 | **59.61** ±8.75 | 112.70 ±28.49 |
| MESMO | SNW | 89.23 ±6.62 | 92.7 ±3.27 | 45.84 ±0.12 | 88.40 ±1.40 | 56.54 ±7.89 | 53.77 ±5.33 | 112 |
| EHVI | SNW | 26.54 ±16.80 | 46.62 ±25.92 | 32.09 ±5.10 | 72.57 ±12.83 | 16.92 ±13.01 | 14.25 ±13.07 | 112 |
| JESMO | SNW | 0.00 ±0.00 | 1.43 ±4.29 | 27.48 ±1.27 | 85.87 ±0.91 | 0.00 ±0.00 | 0.00 ±0.00 | 112 |
| PESMO | SNW | 0.00 ±0.00 | 0.00 ±0.00 | 26.34 ±2.15 | 86.12 ±0.73 | 0.00 ±0.00 | 0.00 ±0.00 | 112 |
| VOGP (Ours) | BC | **98.33** ±5.00 | **100.0** ±0.00 | **42.66** ±0.06 | **99.20** ±0.59 | **80.00** ±20.82 | **88.33** ±14.53 | 33.6 ±2.73 |
| MESMO | BC | 78.33 ±29.86 | 94.24 ±8.94 | 42.59 ±0.14 | 97.36 ±1.78 | 65.00 ±28.33 | 50.12 ±24.41 | 33 |
| EHVI | BC | 36.67 ±22.11 | 90.67 ±19.6 | 42.14 ±0.29 | 97.08 ±0.74 | 31.67 ±8.98 | 40.02 ±16.70 | 33 |
| JESMO | BC | 46.67 ±26.67 | **100.0** ±0.00 | 42.41 ±0.09 | 96.92 ±0.44 | 36.67 ±6.67 | 36.57 ±7.79 | 33 |
| PESMO | BC | 60.00 ±32.66 | **100.0** ±0.00 | 42.45 ±0.12 | 96.92 ±0.54 | 41.67 ±11.18 | 37.89 ±7.62 | 33 |
| VOGP (Ours) | OKA | 87.14 ±4.29 | 99.09 ±2.73 | 51.68 ±0.02 | 96.24 ±0.57 | 57.14 ±6.39 | 38.91 ±5.84 | 110.60 ±16.38 |
| MESMO | OKA | 77.14 ±7.0 | 98.89 ±3.33 | 50.39 ±1.96 | 96.96 ±0.86 | 48.57 ±9.48 | 48.17 ±13.87 | 110 |
| EHVI | OKA | 88.57 ±5.71 | **100.0** ±0.00 | **51.72** ±0.03 | **97.76** ±0.82 | **65.71** ±9.48 | **60.85** ±14.92 | 110 |
| JESMO | OKA | **90.00** ±6.55 | **100.0** ±0.00 | 51.70 ±0.03 | 97.28 ±0.5 | 62.86 ±7.00 | 51.66 ±7.27 | 110 |
| PESMO | OKA | 87.14 ±7.69 | **100.0** ±0.00 | 51.62 ±0.20 | 96.96 ±0.65 | 61.43 ±9.15 | 47.50 ±9.88 | 110 |
| VOGP (Ours) | SnAr | **67.5** ±40.39 | **95.00** ±10.00 | **48.65** ±1.17 | **99.46** ±0.14 | **40.00** ±25.50 | **34.42** ±21.34 | 39.80 ±20.26 |
| MESMO | SnAr | 15.0 ±25.5 | 56.0 ±35.77 | 45.69 ±1.76 | 99.39 ±0.15 | 5.00 ±10.00 | 13.33 ±30.55 | 39 |
| EHVI | SnAr | 0.00 ±0.00 | 0.00 ±0.00 | 14.29 ±0.00 | 99.47 ±0.00 | 0.00 ±0.0 | 0.0 ±0.00 | 39 |
| JESMO | SnAr | 0.00 ±0.00 | 0.00 ±0.00 | 14.29 ±0.00 | 99.47 ±0.00 | 0.00 ±0.0 | 0.0 ±0.00 | 39 |
| PESMO | SnAr | 0.00 ±0.00 | 0.00 ±0.00 | 14.29 ±0.00 | 99.47 ±0.00 | 0.00 ±0.0 | 0.0 ±0.00 | 39 |

**Experimental Setup** VOGP algorithm was run with the same setup as in the main paper's experiments section. As in the case of VOGP, for MOO methods, initially we learn the kernel hyperparameters by training on the entire dataset. This is done to ensure that the algorithms capture the smoothness of the dataset correctly. Data used in this phase is not used to update the GP posterior. All algorithms start the learning phase with an empty set of observations. MOO methods operates in fixed budget setting. To have an accurate comparison between VOGP and MOO algorithms, we set the budget of MOO algorithms based on the average number of evaluations of VOGP in each dataset. All methods used Matérn kernel for OKA dataset and RBF kernel for other datasets. After the optimization loop of MOO methods is finished, their observations are used to train a GP, and the posterior mean values for designs in the design set are used to calculate the predicted Pareto set for these MOO methods.

**Discussion** As seen in Table 3, VOGP consistently ranks among the top-performing methods across datasets and metrics. For BC, SNW and SnAr datasets, VOGP surpasses other methods in every metric. JESMO performs well in the OKA dataset, especially with SR. EHVI performs well in OKA dataset and has the highest Pareto statistics. MESMO generally has decent performances but does not lead as strongly as VOGP in most cases. In conclusion, VOGP is the most consistently high-performing method across the datasets and metrics presented. In real-world datasets (SnAr, SNW), the lead of VOGP over MOO methods is amplified and PESMO and JESMO fail to return good designs.

The superiority of VOGP over other MOO methods can be attributed to multiple factors. First, VOGP guarantees an $(\epsilon, \delta)$-PAC Pareto set to be returned, whereas the compared MOO methods do not have this guarantee. Second, VOGP algorithm does maximum variance reduction-based sampling. In Vakili et al. (2021a), it was shown that tighter than usual confidence bounds hold for sampling methods where noise history is independent of sampling history. For instance, a purely non-adaptive variance reduction-based algorithm satisfies the conditions for the mentioned tight bounds to hold since it samples without looking at the observations. While VOGP performs variance reduction in a similar spirit, it also performs adaptive elimination, which may prevent a design from getting sampled if the design is discarded based on the noise history. Therefore, these tight bounds are not certified for VOGP. However, it is important to observe that at the beginning of the algorithm, the sampling strategy of VOGP is very close to that of a purely non-adaptive variance reduction-based sampling algorithm. Since we empirically scale VOGP's confidence region in the experiments, VOGP might be getting the benefit of (almost) valid tighter confidence regions, thereby improving its performance.

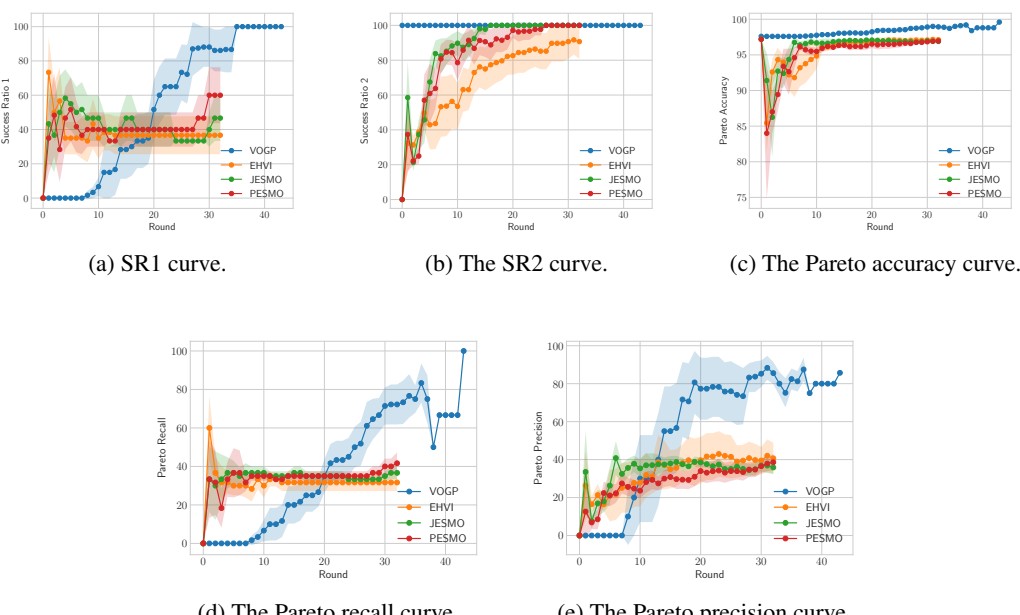

(a) SR1 curve.

(b) The SR2 curve.

(c) The Pareto accuracy curve.

(d) The Pareto recall curve.

(e) The Pareto precision curve.

Figure 3: The scores of VOGP, JESMO, EHVI, and PESMO on BC dataset with $C_{90°}$ cone. Since VOGP's runs are of different round counts, averages were taken over possible runs.

A.2    COMPARISON OF VOGP WITH OTHER MOO METHODS UNDER $C = C_{135°}$

In this section, we compare VOGP run with $C = C_{135°}$ to other MOO methods under with the metrics defined over $C_{135°}$.

Table 4: Comparison of VOGP with other MOO methods under $C_{135°}$. SR1 and SR2: The success rate of satisfying ($\epsilon,\delta$)-PAC conditions ($i$) and ($ii$) respectively. PA: Pareto accuracy rate. PR: Pareto recall rate. PP: Pareto precision rate. SC: The number of evaluations.

| Method | Dataset | SR1 ↑ | SR2 ↑ | PA ↑ | PR ↑ | PP ↑ | SC↓ |
|---|---|---|---|---|---|---|---|
| VOGP (Ours) | SNW | **92.00** ±**9.80** | **94.39** ±**6.41** | **96.12** ±**1.76** | **72.0** ±**12.49** | **60.07** ±**15.58** | 72.8 ±11.91 |
| MESMO | SNW | 80.00 ±14.14 | 80.31 ±5.91 | 94.22 ±1.86 | 63.0 ±20.52 | 43.79 ±11.95 | 72 |
| EHVI | SNW | 0.00 ±0.00 | 50.00 ±50.00 | 94.66 ±0.00 | 0.00 ±0.00 | 0.00 ±0.00 | 72 |
| JESMO | SNW | 0.00 ±0.00 | 0.00 ±0.00 | 94.66 ±0.00 | 0.00 ±0.00 | 0.00 ±0.00 | 72 |
| PESMO | SNW | 0.00 ±0.00 | 0.00 ±0.00 | 94.66 ±0.00 | 0.00 ±0.00 | 0.00 ±0.00 | 72 |
| VOGP (Ours) | BC | **100.00** ±**0.00** | **100.00** ±**0.00** | 99.88 ±0.18 | 85.0 ±22.91 | **100.00** ±**0.00** | 17.5 ±2.77 |
| MESMO | BC | 60.00 ±48.99 | 70.00 ±45.83 | 98.0 ±3.01 | 45.0 ±41.53 | 60.00 ±48.99 | 17 |
| EHVI | BC | **100.00** ±**0.00** | **100.00** ±**0.00** | 99.92 ±0.16 | 90.00 ±20.00 | **100.00** ±**0.00** | 17 |
| JESMO | BC | **100.00** ±**0.00** | **100.00** ±**0.00** | **99.96** ±**0.12** | **95.0** ±**15.0** | **100.00** ±**0.00** | 17 |
| PESMO | BC | **100.00** ±**0.00** | **100.00** ±**0.00** | 99.84 ±0.20 | 80.00 ±24.49 | **100.00** ±**0.00** | 17 |
| VOGP (Ours) | OKA | 65.00 ±22.91 | 85.00 ±22.91 | 99.60 ±0.31 | 65.00 ±22.91 | 85.00 ±22.91 | 44.60 ±5.22 |
| MESMO | OKA | 50.00 ±0.00 | 85.0 ±22.91 | 99.48 ±0.18 | 50.00 ±0.00 | 85.00 ±22.91 | 44 |
| EHVI | OKA | 60.00 ±20.00 | **100.00** ±**0.00** | 99.68 ±0.16 | 60.00 ±20.00 | **100.00** ±**0.00** | 44 |
| JESMO | OKA | **80.00** ±**24.49** | 96.67 ±10.00 | **99.80** ±**0.20** | **80.00** ±**24.49** | 96.67 ±10.00 | 44 |
| PESMO | OKA | 70.00 ±24.49 | 96.67 ±10.00 | 99.72 ±0.18 | 70.00 ±24.49 | 96.67 ±10.00 | 44 |
| VOGP (Ours) | SnAr | **73.33** ±**13.33** | **100.00** ±**0.00** | **99.64** ±**0.1** | **26.67** ±**20.00** | **38.33** ±**30.78** | 32.9 ±9.41 |
| MESMO | SnAr | 30.00 ±31.45 | 60.00 ±48.99 | 99.61 ±0.09 | 6.67 ±13.33 | 20.00 ±40.00 | 32 |
| EHVI | SnAr | 0.00 ±0.00 | 0.00 ±0.00 | 99.58 ±0.00 | 0.00 ±0.00 | 0.00 ±0.00 | 32 |
| JESMO | SnAr | 0.00 ±0.00 | 0.00 ±0.00 | 99.58 ±0.00 | 0.00 ±0.00 | 0.00 ±0.00 | 32 |
| PESMO | SnAr | 0.00 ±0.00 | 0.00 ±0.00 | 99.58 ±0.00 | 0.00 ±0.00 | 0.00 ±0.00 | 32 |

**Experimental Setup** In this experiment (by all methods), for OKA dataset, Matérn kernel was used. For SNW, BC, SnAr datasets, RBF kernel was used. Gaussian process kernel parameters are known by all methods. MOO methods are run with the budget of VOGP for $C = C_{135°}$. After MOO methods finish the Bayseian optimization loop, a GP is set with the observations of the MOO methods. Then, the posterior mean values of the GP are used (by identifying the non-dominated posterior means) to calculate the estimated Pareto set for $C_{135°}$. PA, PR, PP are then calculated with respect to $C_{135°}$.

**Discussion** VOGP consistently ranks among the top-performing methods across datasets and metrics. While VOGP consistently demonstrates high success rates (SR1 and SR2), other methods show a broader range of performance. This result is not unexpected since VOGP performs comparisons with the ordering cone whereas for the other methods, there is a mismatch between their optimized acquisition function and the cone-dependent Pareto front.

B    COMPARISON OF VOGP WITH RANDOM SEARCH

Here, we compare the VOGP algorithm with random search as a baseline. The random search algorithm samples as many samples as VOGP had sampled at random. Then, the observed data is used to train a Gaussian process model, whose posterior mean values at every design are used to determine the cone-dependent Pareto set.

Table 5: Performance comparison between VOGP (Our method) and random search under $C_{45°}$, $C_{90°}$, and $C_{135°}$. SR1 and SR2: The success rate of satisfying $(\epsilon,\delta)$-PAC conditions $(i)$ and $(ii)$ respectively. PA: Pareto accuracy rate. PR: Pareto recall rate. PP: Pareto precision rate. SC: The number of evaluations.

| | | VOGP | | | | | | Random search | | | | | |
|---|---|---|---|---|---|---|---|---|---|---|---|---|---|
| $C$ | D | SR1↑ | SR2↑ | PA↑ | PR↑ | PP↑ | SC | SR1↑ | SR2↑ | PA↑ | PR↑ | PP↑ | SC |
| $C_{45°}$ | SNW | 95.96 ±2.78 | 98.65 ±1.46 | 85.53 ±1.35 | 76.54 ±4.93 | 69.49 ±3.25 | 777.8 ±220.65 | 95.38 ±0.94 | 99.24 ±0.94 | 85.63 ±1.52 | 73.27 ±4.07 | 70.87 ±3.01 | 777 |
| $C_{45°}$ | BC | 95.17 ±7.43 | 99.37 ±1.27 | 95.36 ±1.59 | 86.90 ±8.13 | 76.56 ±6.35 | 446.40 ±131.14 | 99.66 ±1.03 | 100.00 ±0.00 | 97.24 ±0.9 | 91.03 ±4.14 | 86.17 ±4.71 | 446 |
| $C_{45°}$ | OKA | 88.00 ±7.38 | 92.84 ±5.41 | 95.28 ±0.99 | 76.40 ±4.54 | 76.87 ±6.89 | 650.3 ±153.36 | 97.6 ±4.80 | 94.82 ±5.31 | 95.6 ±0.88 | 79.60 ±6.56 | 77.63 ±5.98 | 650 |
| $C_{45°}$ | SnAr | 78.75 ±14.03 | 98.00 ±6.00 | 98.46 ±0.42 | 55.62 ±16.64 | 53.63 ±12.25 | 517.0 ±148.79 | 87.50 ±7.40 | 96.89 ±5.63 | 98.53 ±0.33 | 53.12 ±11.61 | 57.09 ±12.1 | 517 |
| $C_{90°}$ | SNW | 94.62 ±4.28 | 97.5 ±2.22 | 89.85 ±2.14 | 63.85 ±10.35 | 59.61 ±8.75 | 112.70 ±28.49 | 85.38 ±7.65 | 93.94 ±3.29 | 87.72 ±3.32 | 50.00 ±13.76 | 51.75 ±13.39 | 112 |
| $C_{90°}$ | BC | 98.33 ±5.00 | 100.0 ±0.00 | 99.20 ±0.59 | 80.00 ±20.82 | 88.33 ±14.53 | 33.60 ±2.73 | 98.33 ±5.00 | 88.09 ±11.26 | 98.00 ±0.91 | 88.33 ±13.02 | 59.57 ±17.02 | 33 |
| $C_{90°}$ | OKA | 87.14 ±4.29 | 99.09 ±2.73 | 96.24 ±0.57 | 57.14 ±6.39 | 38.91 ±5.84 | 110.60 ±16.38 | 72.86 ±10.00 | 87.37 ±7.85 | 95.52 ±0.59 | 37.14 ±9.48 | 27.96 ±6.62 | 110 |
| $C_{90°}$ | SnAr | 67.5 ±40.39 | 95.00 ±10.00 | 99.46 ±0.14 | 40.00 ±25.50 | 34.42 ±21.34 | 39.80 ±20.26 | 62.50 ±16.77 | 84.58 ±19.37 | 99.45 ±0.28 | 20.00 ±18.71 | 38.33 ±37.31 | 39 |
| $C_{90°}^{4}$ | MAR | 100.00 ±0.00 | 99.67 ±0.81 | 97.70 ±0.31 | 79.32 ±2.96 | 66.84 ±4.29 | 716.43 ±173.89 | 98.16 ±2.37 | 100.00 ±0.00 | 97.96 ±0.53 | 73.42 ±6.17 | 74.33 ±9.78 | 716 |
| $C_{135°}$ | SNW | 92.00 ±9.80 | 94.39 ±6.41 | 96.12 ±1.76 | 72.00 ±12.49 | 60.07 ±15.58 | 72.80 ±11.91 | 76.00 ±16.85 | 91.79 ±9.27 | 95.10 ±2.12 | 57.00 ±21.00 | 50.50 ±17.86 | 72 |
| $C_{135°}$ | BC | 100.00 ±0.00 | 100.00 ±0.00 | 99.88 ±0.18 | 85.00 ±22.91 | 100.0 0 | 17.50 ±2.77 | 75.00 ±44.73 | 84.68 ±25.51 | 98.84 ±1.49 | 70.00 ±40.00 | 55.77 ±40.57 | 17 |
| $C_{135°}$ | OKA | 65.00 ±22.91 | 85.00 ±22.91 | 99.60 ±0.31 | 65.00 ±22.91 | 85.00 ±22.91 | 44.60 ±5.22 | 30.00 ±24.49 | 33.33 ±37.08 | 98.76 ±0.49 | 30.00 ±24.49 | 33.33 ±37.08 | 44 |
| $C_{135°}$ | SnAr | 73.33 ±13.33 | 100.00 ±0.00 | 99.64 ±0.10 | 26.67 ±20.00 | 38.33 ±30.78 | 32.90 ±9.41 | 63.33 ±23.33 | 66.67 ±37.37 | 99.54 ±0.14 | 13.33 ±16.33 | 20.00 ±31.45 | 32 |

**Experimental Setup** The setup of this experiment is the same as the experiment setup in Section 5.

**Discussion** As seen in Table 5, VOGP consistently outperforms random search method across datasets and metrics for cones $C_{90°}$ and $C_{135°}$. The superiority of VOGP stems from its efficient sampling as a result of variance reduction sampling. By learning as much as possible with few samples, GP used in VOGP learns more about the underlying objective function than random search. For $C_{45°}$, random search method performs marginally better than VOGP. This could be because VOGP samples more designs than are necessary to learn the objective function globally, possibly because of looseness in confidence intervals of designs. This makes the advantages of variance reduction of VOGP diminish, making random search learn the function well enough to become competitive with VOGP.

# C    ABLATION STUDY: UNKNOWN KERNEL HYPERPARAMETERS

In numerous practical situations, the correct hyperparameters are often inaccessible to us, nor do we possess reliable initial estimates for them. Therefore, a common approach in practical settings is to gradually learn and adjust the hyperparameters as more queries get included in observations. In this section, we investigate the performance of VOGP when the kernel hyperparameters are unknown. Then, as more observations are made, at each round, the kernel hyperparameters are retrained with the observed data so far. We also share the results of an experiment where VOGP's parameters are fixed after being trained with a small sample of 30 datapoints. In the latter setting, parameters are not updated as more queries get included in observations.

Table 6: Performance analysis of VOGP (Our method) with known and unknown kernel hyperparameters under $C_{45°}$, $C_{90°}$, and $C_{135°}$. SR1 and SR2: The success rate of satisfying $(\epsilon,\delta)$-PAC conditions $(i)$ and $(ii)$ respectively. PA: Pareto accuracy rate. PR: Pareto recall rate. PP: Pareto precision rate. SC: The number of evaluations.

| | | VOGP (Known Parameters) | | | | | | VOGP (Unknown Parameters) | | | | | |
|---|---|---|---|---|---|---|---|---|---|---|---|---|---|
| $C$ | D | SR1↑ | SR2↑ | PA↑ | PR↑ | PP↑ | SC | SR1↑ | SR2↑ | PA↑ | PR↑ | PP↑ | SC |
| $C_{45°}$ | SNW | 95.96 ±2.78 | 98.65 ±1.46 | 85.53 ±1.35 | 76.54 ±4.93 | 69.49 ±3.25 | 777.80 ±220.65 | 98.46 ±2.24 | 86.93 ±17.95 | 72.33 ±23.71 | 83.46 ±10.29 | 57.74 ±17.08 | 283.2 ±206.97 |
| $C_{45°}$ | BC | 95.17 ±7.43 | 99.37 ±1.27 | 95.36 ±1.59 | 86.90 ±8.13 | 76.56 ±6.35 | 446.40 ±131.14 | 70.34 ±35.21 | 81.1 ±31.33 | 90.84 ±3.62 | 57.24 ±31.35 | 52.46 ±25.14 | 180.8 ±102.00 |
| $C_{45°}$ | OKA | 88.00 ±7.38 | 92.84 ±5.41 | 95.28 ±0.99 | 76.40 ±4.54 | 76.87 ±6.89 | 650.3 ±153.36 | 64.50 ±27.82 | 61.42 ±29.96 | 80.9 ±26.83 | 59.0 ±27.69 | 42.83 ±22.63 | 254.25 ±179.59 |
| $C_{45°}$ | SnAr | 78.75 ±14.03 | 98.00 ±6.00 | 98.46 ±0.42 | 55.62 ±16.64 | 53.63 ±12.25 | 517.00 ±148.79 | 73.75 ±29.82 | 66.54 ±34.44 | 78.96 ±38.64 | 55.62 ±32.05 | 36.38 ±22.1 | 93.9 ±64.71 |
| $C_{90°}$ | SNW | 94.62 ±4.28 | 97.5 ±2.22 | 89.85 ±2.14 | 63.85 ±10.35 | 59.61 ±8.75 | 112.70 ±28.49 | 73.85 ±37.0 | 72.51 ±36.46 | 86.99 ±2.26 | 53.85 ±27.41 | 39.53 ±20.75 | 104.4 ±86.2 |
| $C_{90°}$ | BC | 98.33 ±5.00 | 100.0 ±0.00 | 99.20 ±0.59 | 80.00 ±20.82 | 88.33 ±14.53 | 33.60 ±2.73 | 30.00 ±38.59 | 73.41 ±31.02 | 87.24 ±28.3 | 28.33 ±38.04 | 15.57 ±21.56 | 15.0 ±12.75 |
| $C_{90°}$ | OKA | 87.14 ±4.29 | 99.09 ±2.73 | 96.24 ±0.57 | 57.14 ±6.39 | 38.91 ±5.84 | 110.60 ±16.38 | 69.64 ±29.83 | 50.17 ±39.29 | 73.4 ±38.21 | 46.43 ±34.07 | 24.17 ±24.81 | 29.88 ±21.07 |
| $C_{90°}$ | SnAr | 67.5 ±40.39 | 95.00 ±10.00 | 99.46 ±0.14 | 40.00 ±25.50 | 34.42 ±21.34 | 39.80 ±20.26 | 17.5 ±35.44 | 33.57 ±39.75 | 99.24 ±0.21 | 10.00 ±20.00 | 6.86 ±13.95 | 19.7 ±11.47 |
| $C_{135°}$ | SNW | 92.00 ±9.80 | 94.39 ±6.41 | 96.12 ±1.76 | 72.00 ±12.49 | 60.07 ±15.58 | 72.80 ±11.91 | 64.00 ±34.99 | 70.28 ±37.72 | 94.76 ±0.84 | 56.00 ±31.37 | 38.17 ±19.84 | 67.6 ±42.26 |
| $C_{135°}$ | BC | 100.00 ±0.00 | 100.00 ±0.00 | 99.88 ±0.18 | 85.00 ±22.91 | 100.00 ±0.00 | 17.50 ±2.77 | 60.00 ±48.99 | 51.91 ±44.73 | 88.8 ±29.37 | 60.0 ±48.99 | 35.08 ±36.78 | 13.6 ±8.24 |
| $C_{135°}$ | OKA | 65.00 ±22.91 | 85.00 ±22.91 | 99.60 ±0.31 | 65.00 ±22.91 | 85.00 ±22.91 | 44.60 ±5.22 | 18.75 ±34.8 | 6.35 ±16.5 | 86.20 ±32.28 | 18.75 ±34.8 | 6.35 ±16.5 | 13.75 ±12.4 |
| $C_{135°}$ | SnAr | 73.33 ±13.33 | 100.00 ±0.00 | 99.64 ±0.10 | 26.67 ±20.00 | 38.33 ±30.78 | 32.90 ±9.41 | 10.00 ±21.34 | 25.00 ±40.31 | 99.58 ±0.12 | 10.00 ±21.34 | 15.00 ±32.02 | 12.00 ±5.78 |

**Experimental Setup** the In this experiment (by all methods), for OKA dataset, Matérn kernel was used. For SNW, BC, SnAr datasets, RBF kernel was used. The instance of VOGP with the unknown kernel hyperparameters, as more observations are made at each round, are retrained with the observed data so far.

Table 7: Performance analysis of VOGP (Our method) with known and partially known hyperparameters under $C_{45°}$, $C_{90°}$, and $C_{135°}$. SR1 and SR2: The success rate of satisfying $(\epsilon,\delta)$-PAC conditions $(i)$ and $(ii)$ respectively. PA: Pareto accuracy rate. PR: Pareto recall rate. PP: Pareto precision rate. SC: The number of evaluations.

| | | VOGP (Known Parameters) | | | | | | VOGP (Partially Known Parameters) | | | | | |
|---|---|---|---|---|---|---|---|---|---|---|---|---|---|
| $C$ | D | SR1 ↑ | SR2 ↑ | PA ↑ | PR ↑ | PP ↑ | SC ↓ | SR1 ↑ | SR2 ↑ | PA ↑ | PR ↑ | PP ↑ | SC ↓ |
| $C_{45°}$ | SNW | 95.96 ±2.78 | 98.65 ±1.46 | 85.53 ±1.35 | 76.54 ±4.93 | 69.49 ±3.25 | 777.8 ±220.65 | 96.35 ±2.50 | 97.83 ±1.73 | 82.67 ±2.35 | 68.08 ±5.52 | 65.28 ±5.18 | 450.7 ±117.68 |
| $C_{45°}$ | BC | 95.17 ±7.43 | 99.37 ±1.27 | 95.36 ±1.59 | 86.90 ±8.13 | 76.56 ±6.35 | 446.40 ±131.14 | 92.07 ±5.78 | 96.65 ±3.50 | 92.48 ±2.54 | 78.28 ±7.24 | 65.7 ±10.44 | 285.3 ±93.17 |
| $C_{45°}$ | OKA | 88.00 ±7.38 | 92.84 ±5.41 | 95.28 ±0.99 | 76.40 ±4.54 | 76.87 ±6.89 | 650.3 ±153.36 | 82.0 ±7.43 | 89.7 ±2.53 | 93.16 ±1.31 | 74.0 ±7.21 | 63.78 ±6.0 | 365.2 ±154.56 |
| $C_{45°}$ | SnAr | 78.75 ±14.03 | 98.00 ±6.00 | 98.46 ±0.42 | 55.62 ±16.64 | 53.63 ±12.25 | 517.0 ±148.79 | 90.0 ±9.76 | 96.81 ±4.31 | 98.72 ±0.31 | 60.0 ±8.93 | 62.82 ±9.74 | 350.8 ±91.41 |
| $C_{90°}$ | SNW | 94.62 ±4.28 | 97.5 ±2.22 | 89.85 ±2.14 | 63.85 ±10.35 | 59.61 ±8.75 | 112.70 ±28.49 | 98.08 ±3.10 | 93.23 ±3.60 | 89.17 ±1.55 | 64.62 ±11.12 | 56.36 ±5.29 | 69.5 ±13.29 |
| $C_{90°}$ | BC | 98.33 ±5.00 | 100.0 ±0.00 | 99.20 ±0.59 | 80.00 ±20.82 | 88.33 ±14.53 | 33.60 ±2.73 | 93.33 ±20.0 | 98.89 ±3.33 | 98.96 ±0.93 | 80.0 ±19.44 | 79.89 ±21.36 | 28.5 ±8.02 |
| $C_{90°}$ | OKA | 87.14 ±4.29 | 99.09 ±2.73 | 96.24 ±0.57 | 57.14 ±6.39 | 38.91 ±5.84 | 110.60 ±16.38 | 75.71 ±6.55 | 90.44 ±10.64 | 96.28 ±0.67 | 40.0 ±12.45 | 35.66 ±9.93 | 52.5 ±15.31 |
| $C_{90°}$ | SnAr | 67.5 ±40.39 | 95.00 ±10.00 | 99.46 ±0.14 | 40.00 ±25.50 | 34.42 ±21.34 | 39.80 ±20.26 | 82.5 ±11.46 | 82.92 ±22.55 | 99.51 ±0.16 | 37.5 ±16.77 | 51.0 ±26.78 | 27.0 ±11.08 |
| $C_{135°}$ | SNW | 92.00 ±9.80 | 94.39 ±6.41 | 96.12 ±1.76 | 72.00 ±12.49 | 60.07 ±15.58 | 72.80 ±11.91 | 95.00 ±8.06 | 90.06 ±10.99 | 96.65 ±1.56 | 85.0 ±8.06 | 64.18 ±15.87 | 42.0 ±7.21 |
| $C_{135°}$ | BC | 100.00 ±0.00 | 100.00 ±0.00 | 99.88 ±0.18 | 85.00 ±22.91 | 100.00 ±0.00 | 17.50 ±2.77 | 100.0 ±0.0 | 100.0 ±0.0 | 100.0 ±0.0 | 100.0 ±0.0 | 100.0 ±0.0 | 13.3 ±1.95 |
| $C_{135°}$ | OKA | 65.00 ±22.91 | 85.00 ±22.91 | 99.60 ±0.31 | 65.00 ±22.91 | 85.00 ±22.91 | 44.60 ±5.22 | 40.0 ±20.0 | 55.0 ±35.0 | 99.08 ±0.57 | 40.0 ±20.0 | 55.0 ±35.0 | 25.9 ±6.86 |
| $C_{135°}$ | SnAr | 73.33 ±13.33 | 100.00 ±0.00 | 99.64 ±0.10 | 26.67 ±20.00 | 38.33 ±30.78 | 32.90 ±9.41 | 66.67 ±29.81 | 75.0 ±33.54 | 99.68 ±0.19 | 36.67 ±17.95 | 60.0 ±30.0 | 16.7 ±4.08 |

**Experimental Setup** In this experiment, for OKA dataset, Matérn kernel was used. For SNW, BC, SnAr datasets, RBF kernel was used. The instance of VOGP with the partially known kernel hyperparameters has its parameters with randomly selected 30 datapoints initially. These data are not used to update the GP posterior.

**Discussion** As seen in Table 6, VOGP's performance is affected negatively when the kernel parameters are initially unknown. Though the VOGP gets to learn the kernel parameters as more queries are added to the observations, VOGP performs Pareto classification operations in early stages without having the right kernel parameters. Since VOGP does not check discarded designs again and does not reconsider added Pareto designs, these erroneous decisions in the cannot be corrected later on. As seen in Table 7, a small set of 30 initial samples to learn the kernel parameters are enough to make VOGP perform marginally worse than the version with known parameters. Additionally, VOGP samples the objective function less times when it has incomplete information of the kernel hyperparameters. A possible reason for this would be higher length scales, which could lead to observations reducing more uncertainty about other designs, leading to faster terminations.

## D    RELATED WORKS ON EVOLUTIONARY ALGORITHMS

Numerous studies employ multi-objective evolutionary algorithms in order to estimate the Pareto front, thereby accomplishing this task through the iterative development of a population of evaluated designs; see Seada & Deb (2015); Deb et al. (2002); Knowles (2006); Zhang & Li (2007). Some of these methods use hypervolume calculation to guide their method (Yang et al., 2016b; Beume et al., 2007; Bader & Zitzler, 2011). A significant body of research exists on the integration of user preferences into multi-objective evolutionary algorithms (Coello, 2000; Zhou et al., 2011; Branke & Deb, 2005; Phelps & Köksalan, 2003; Li & Silva, 2008; Sinha et al., 2010). Using evolutionary algorithms, preferences can be articulated *a priori*, *a posteriori*, or interactively. Various methods for expressing preferences have been suggested, primarily encompassing techniques based on objective comparisons, solution ranking, and expectation-based approaches (Zhou et al., 2011). There are also works that employ preference cones in multi-objective evolutionary algorithms. Batista et al. (2011) utilize polyhedral cones as a method of managing the resolution of the estimated Pareto front. Ferreira et al. (2020) apply preference cone based multi-objective evolutionary algorithm to optimize distributed energy resources in microgrids.

# E   PROOFS

Below, we present a table of symbols that are used in the proofs.

| Symbol | Description |
| --- | --- |
| $\mathcal{X}$ | The design space |
| $M$ | The dimension of the objective space |
| $\mathbb{S}^{M-1}$ | The unit sphere in $\mathbb{R}^M$ |
| $C$ | The polyhedral ordering cone |
| $\zeta$ | The ordering complexity of the cone $C$ |
| $\boldsymbol{f}$ | The objective function |
| $\epsilon$ | Accuracy level given as input to the algorithm |
| $\boldsymbol{y}_{[t]}$ | Vector that represents the first $t$ noisy observations where $\boldsymbol{y}_{[0]} = \emptyset$. |
| $\boldsymbol{\mu}_t(x)$ | The posterior mean of design $x$ at round $t$ whose $j^{\text{th}}$ component is $\mu_t^j(x)$ |
| $\boldsymbol{\sigma}_t(x)$ | The posterior variance of design $x$ at round $t$ whose $j^{\text{th}}$ component is $\sigma_t^j(x)$ |
| $\beta_t$ | The confidence term at round $t$ |
| $\mathcal{P}_t$ | The predicted Pareto set of designs at round $t$ |
| $\mathcal{S}_t$ | The undecided sets of designs at round $t$ |
| $\hat{\mathcal{P}}$ | The estimated Pareto set of designs returned by VOGP |
| $P^*$ | The set of true Pareto optimal designs |
| $P_\theta^*$ | The set of true Pareto optimal designs when $M = 2$ and $C = C_\theta$ |
| $\mathcal{A}_t$ | The union of sets $\mathcal{S}_t$ and $\mathcal{P}_t$ at the beginning of round $t$ |
| $\mathcal{W}_t$ | The union of sets $\mathcal{S}_t$ and $\mathcal{P}_t$ at the end of the discarding phase of round $t$ |
| $\boldsymbol{Q}_t(x)$ | The confidence hyperrectangle associated with design $x$ at round $t$ |
| $\boldsymbol{R}_t(x)$ | The cumulative confidence hyperrectangle associated with design $x$ at round $t$ |
| $x_t$ | The design evaluated at round $t$ |
| $\omega_t(x)$ | The diameter of the cumulative confidence hyperrectangle of design $x$ at round $t$ |
| $\overline{\omega}_t$ | The maximum value of $\omega_t(x)$ over all active designs $x$ at round $t$ |
| $m(x, x')$ | $\inf\{s \geq 0 \mid \exists \boldsymbol{u} \in \mathbb{B}(1) \cap C : \boldsymbol{f}(x) + s\boldsymbol{u} \notin \boldsymbol{f}(x') - \text{int}(C)\}$ |
| $\gamma_t$ | The maximum information that can be gained about $\boldsymbol{f}$ in $t$ evaluations |
| $t_s$ | The round in which VOGP terminates |

### E.1 DERIVATION OF THEOREM 1

Let $\epsilon > 0$ and $\delta \in (0, 1)$ be given. Let $\mathbb{N} := \{1, 2, \ldots\}$.

**Lemma 2.** *Let us define the event*

$$E := \left\{ \forall j \in [M] \; \forall t \in \mathbb{N} \; \forall x \in \mathcal{X} \colon |f^j(x) - \mu_t^j(x)| \leq \beta_t^{1/2} \sigma_t^j(x) \right\},$$

*where*

$$\beta_t := \ln\left( \frac{M \pi^2 |\mathcal{X}| t^2}{3\delta} \right).$$

*Then, $\mathbb{P}(E) \geq 1 - \delta$.*

*Proof.* For an event $E'$, we denote by $\mathbb{I}(E')$ its probabilistic indicator function, i.e., $\{\mathbb{I}(E') = 1\} = E'$ and $\{\mathbb{I}(E') = 0\} = \Omega \setminus E'$, where $\Omega$ is the underlying sample space. Note that

$$1 - \mathbb{P}(E) = \mathbb{E}\left[ \mathbb{I}\left( \left\{ \exists j \in [M] \; \exists t \in \mathbb{N} \; \exists x \in \mathcal{X} \colon |f^j(x) - \mu_t^j(x)| > \beta_t^{1/2} \sigma_t^j(x) \right\} \right) \right]$$

$$\leq \mathbb{E}\left[ \sum_{j=1}^{M} \sum_{t=1}^{\infty} \sum_{x \in \mathcal{X}} \mathbb{I}\left( \left\{ |f^j(x) - \mu_t^j(x)| > \beta_t^{1/2} \sigma_t^j(x) \right\} \right) \right]$$

$$= \sum_{j=1}^{M} \sum_{t=1}^{\infty} \sum_{x \in \mathcal{X}} \mathbb{E}\left[ \mathbb{E}\left[ \mathbb{I}\left( \left\{ |f^j(x) - \mu_t^j(x)| > \beta_t^{1/2} \sigma_t^j(x) \right\} \right) \Big| \boldsymbol{y}_{[t-1]} \right] \right] \quad (4)$$

$$= \sum_{j=1}^{M} \sum_{t=1}^{\infty} \sum_{x \in \mathcal{X}} \mathbb{E}\left[ \mathbb{P}\left( \left\{ |f^j(x) - \mu_t^j(x)| > \beta_t^{1/2} \sigma_t^j(x) \right\} \Big| \boldsymbol{y}_{[t-1]} \right) \right]$$

$$\leq \sum_{j=1}^{M} \sum_{t=1}^{\infty} \sum_{x \in \mathcal{X}} 2 e^{-\beta_t/2} \quad (5)$$

$$= 2M|\mathcal{X}| \sum_{t=1}^{\infty} e^{-\beta_t/2}$$

$$= 2M|\mathcal{X}| \sum_{t=1}^{\infty} \left( \frac{M \pi^2 |\mathcal{X}| t^2}{3\delta} \right)^{-1}$$

$$= \frac{6\delta}{\pi^2} \sum_{t=1}^{\infty} \frac{1}{t^2} = \delta,$$

where (4) uses the tower rule and linearity of expectation and (5) uses Gaussian tail bound; here note that, given $\boldsymbol{y}_{[t-1]}$, the conditional distribution of $f^j(x)$ is $\mathcal{N}(\mu_t^j(x), \sigma_t^j(x))$. $\qquad \square$

For each $s > 0$, let us introduce

$$A(s) := \bigcap_{\boldsymbol{u} \in \mathbb{S}^{M-1}} (s\boldsymbol{u} + C), \quad d(s) := \inf\{\|\boldsymbol{z}\|_2 \mid \boldsymbol{z} \in A(s)\}.$$

**Lemma 3.** *Let $s > 0$. Then, $A(s) = sA(1) := \{s\boldsymbol{y} \mid \boldsymbol{y} \in A(1)\}$ and $d(s) = sd(1)$. Moreover, there exists a unique $\boldsymbol{z}^s \in A(s)$ such that $d(s) = \|\boldsymbol{z}^s\|_2$.*

*Proof.* Let $s > 0$. Then, note that

$$\boldsymbol{y} \in A(s) \Leftrightarrow \boldsymbol{y} \in s\boldsymbol{u} + C, \; \forall \boldsymbol{u} \in \mathbb{S}^{M-1}$$

$$\Leftrightarrow \boldsymbol{y} - s\boldsymbol{u} \in C, \; \forall \boldsymbol{u} \in \mathbb{S}^{M-1}$$

$$\Leftrightarrow \boldsymbol{w}_n^\top (\boldsymbol{y} - s\boldsymbol{u}) \geq 0, \; \forall n \in [N], \; \forall \boldsymbol{u} \in \mathbb{S}^{M-1}$$

$$\Leftrightarrow \boldsymbol{w}_n^\top \boldsymbol{y} \geq s \boldsymbol{w}_n^\top \boldsymbol{u}, \; \forall n \in [N], \; \forall \boldsymbol{u} \in \mathbb{S}^{M-1}$$

$$\Leftrightarrow \boldsymbol{w}_n^\top \boldsymbol{y} \geq s \sup_{\boldsymbol{u} \in \mathbb{S}^{M-1}} \left( \boldsymbol{w}_n^\top \boldsymbol{u} \right), \; \forall n \in [N]. \quad (6)$$

Let $n \in [N]$. By the definition of $\boldsymbol{w}_n$, we have

$$\sup_{\boldsymbol{u} \in \mathbb{S}^{M-1}} \boldsymbol{w}_n^\top \boldsymbol{u} = \sup_{\boldsymbol{u} \in \mathbb{B}(1)} \boldsymbol{w}_n^\top \boldsymbol{u} = ||\boldsymbol{w}_n||_2 = 1 . \tag{7}$$

Combining (6) and (7), we get

$$\boldsymbol{y} \in A(s) \Leftrightarrow \boldsymbol{w}_n^\top \boldsymbol{y} \geq s, \ \forall n \in [N] .$$

Therefore, $A(s) = \{\boldsymbol{z} \in \mathbb{R}^M \mid \boldsymbol{w}_n^\top \boldsymbol{z} \geq s, \ \forall n \in [N]\}$, which implies that $A(s) = sA(1)$ and hence $d(s) = sd(1)$.

The existence and uniqueness of $\boldsymbol{z}^s$ is a direct consequence of the strict convexity and continuity of the $\ell_2$-norm $\|\cdot\|_2$ together with the closedness and convexity of $A(s)$. $\qquad\square$

**Remark 3.** *The last part of Lemma 3 justifies the definition of $\boldsymbol{u}^*$ (see Definition 4).*

In what follows, we denote by $\mathbb{S}^{M-1}(r)$ the boundary of $\mathbb{B}(r)$, where $r > 0$; in particular, $\mathbb{S}^{M-1}(1) = \mathbb{S}^{M-1}$ is the unit sphere.

We next prove Lemma 1: For every $\boldsymbol{y}, \boldsymbol{z} \in \mathbb{R}^M$ and $\tilde{\boldsymbol{p}} \in \mathbb{B}(\frac{\epsilon}{d(1)})$, having $\boldsymbol{y} + \tilde{\boldsymbol{p}} \preccurlyeq_C \boldsymbol{z}$ implies $\boldsymbol{y} \preccurlyeq_C \boldsymbol{z} + \epsilon\boldsymbol{u}^*$.

*Proof of Lemma 1.* Since $\tilde{\boldsymbol{p}} \in \mathbb{B}\left(\frac{\epsilon}{d(1)}\right)$, we have

$$(\tilde{\boldsymbol{p}} + C) \cap \mathbb{S}^{M-1}\left(\frac{\epsilon}{d(1)}\right) \neq \emptyset \iff \exists \tilde{\boldsymbol{\gamma}} \in \mathbb{S}^{M-1}\left(\frac{\epsilon}{d(1)}\right) : \tilde{\boldsymbol{p}} \preccurlyeq_C \tilde{\boldsymbol{\gamma}} . \tag{8}$$

By Definition 4, we have

$$\boldsymbol{u}^* = \frac{\operatorname*{argmin}\limits_{\boldsymbol{z} \in A(1)}(\|\boldsymbol{z}\|_2)}{d(1)} \implies d(1)\boldsymbol{u}^* \in A(1)$$

$$\implies \boldsymbol{u}^* \in \frac{A(1)}{d(1)} \tag{9}$$

$$\implies \boldsymbol{u}^* \in A\left(\frac{1}{d(1)}\right) \tag{10}$$

$$\implies \epsilon\boldsymbol{u}^* \in A\left(\frac{\epsilon}{d(1)}\right), \tag{11}$$

where (9), (10), and (11) follow from Lemma 3. The division in (10) denotes the scaling of the elements of $A(1)$. The proof is completed by observing that, by Definition 4,

$$\forall \boldsymbol{\gamma} \in \mathbb{S}^{M-1}\left(\frac{\epsilon}{d(1)}\right), \forall \boldsymbol{k} \in A\left(\frac{\epsilon}{d(1)}\right) : \boldsymbol{\gamma} \preccurlyeq_C \boldsymbol{k} \tag{12}$$

$$\implies \tilde{\boldsymbol{\gamma}} \preccurlyeq_C \epsilon\boldsymbol{u}^* \tag{13}$$

$$\implies \tilde{\boldsymbol{p}} \preccurlyeq_C \epsilon\boldsymbol{u}^* , \tag{14}$$

where (13) follows from $\tilde{\boldsymbol{\gamma}} \in \mathbb{S}^{M-1}\left(\frac{\epsilon}{d(1)}\right)$ and (11), and (14) follows from combining (13) with (8). $\qquad\square$

**Lemma 4.** *Under event E, the set $\hat{\mathcal{P}}$ returned by VOGP satisfies condition (i) in Definition 1.*

*Proof.* We claim that for every $x \in P^*$, there exists $z \in \hat{\mathcal{P}}$ such that $\boldsymbol{f}(x) \preccurlyeq_C \boldsymbol{f}(z) + \epsilon\boldsymbol{u}^*$. If $x \in \hat{\mathcal{P}}$, then the claim holds with $z = x$, i.e., $\boldsymbol{f}(x) \preccurlyeq_C \boldsymbol{f}(x) + \epsilon\boldsymbol{u}^*$, since $\epsilon\boldsymbol{u}^* \in C$. If $x \notin \hat{\mathcal{P}}$, then $x$ must have been discarded at some round $s_1$. By the discarding rule, that means there exists $z_1 \in \mathcal{P}_{\text{pess},s_1}$ such that

$$\boldsymbol{R}_{s_1}(z_1) + \epsilon\boldsymbol{u}^* \subseteq \boldsymbol{y} + C . \tag{15}$$

holds for every $\boldsymbol{y} \in \boldsymbol{R}_{s_1}(x)$.

At each round, the initial confidence hyperrectangle of a design $x$ is calculated as

$$\boldsymbol{Q}_t(x) = \left\{\boldsymbol{y} \in \mathbb{R}^M \mid \boldsymbol{\mu}_t(x) - \beta_t^{1/2}\boldsymbol{\sigma}_t(x) \preccurlyeq_{\mathbb{R}_+^M} \boldsymbol{y} \preccurlyeq_{\mathbb{R}_+^M} \boldsymbol{\mu}_t(x) + \beta_t^{1/2}\boldsymbol{\sigma}_t(x)\right\} \tag{16}$$

and the initial confidence hyperrectangle is intersected with previous hyperrectangles to obtain the confidence hyperrectangle of the current round, that is,

$$\boldsymbol{R}_t(x) = \boldsymbol{R}_{t-1}(x) \cap \boldsymbol{Q}_t(x) . \tag{17}$$

It can be checked that, due to Lemma 2, we have $\boldsymbol{f}(x) \in \boldsymbol{R}_{s_1}(x)$ under event $E$. Note that, by (15), there exists $z_1 \in \mathcal{P}_{\text{pess},s_1}$ such that

$$\boldsymbol{R}_{s_1}(z_1) + \epsilon \boldsymbol{u}^* \subseteq \boldsymbol{f}(x) + C . \tag{18}$$

Since $\boldsymbol{f}(z_1) \in \boldsymbol{R}_{s_1}(x)$, (18) implies that $\boldsymbol{f}(z_1) + \epsilon \boldsymbol{u}^* \in \boldsymbol{f}(x) + C$, which is equivalent to $\boldsymbol{f}(x) \preccurlyeq_C \boldsymbol{f}(z_1) + \epsilon \boldsymbol{u}^*$. Therefore, if $z_1 \in \hat{\mathcal{P}}$, then the claim holds by choosing $z = z_1$.

If $z_1 \notin \hat{\mathcal{P}}$, then it must have been discarded at some round $s_2 \geq s_1$. Because VOGP discards from the set $S_t \setminus \mathcal{P}_{\text{pess},t}$ at round $t$, $z_1 \notin \mathcal{P}_{\text{pess},s_2}$. Then, using the definition of the pessimistic set (see Definition 3), there exists $z_2 \in \mathcal{A}_{s_2}$ such that

$$\boldsymbol{R}_{s_2}(z_2) \subseteq \boldsymbol{R}_{s_2}(z_1) + C . \tag{19}$$

To proceed, we use (18) and the fact that $C + C = C$, to obtain

$$\begin{aligned}
\boldsymbol{R}_{s_1}(z_1) + \epsilon \boldsymbol{u}^* \subseteq \boldsymbol{f}(x) + C &\implies \boldsymbol{R}_{s_1}(z_1) + \epsilon \boldsymbol{u}^* + C \subseteq \boldsymbol{f}(x) + C + C \\
&\implies \boldsymbol{R}_{s_1}(z_1) + \epsilon \boldsymbol{u}^* + C \subseteq \boldsymbol{f}(x) + C .
\end{aligned} \tag{20}$$

In addition, (19) implies the following

$$\boldsymbol{R}_{s_2}(z_2) + \epsilon \boldsymbol{u}^* \subseteq \boldsymbol{R}_{s_1}(z_1) + \epsilon \boldsymbol{u}^* + C .$$

Combining the above display with (20) yields

$$\boldsymbol{R}_{s_2}(z_2) + \epsilon \boldsymbol{u}^* \subseteq \boldsymbol{R}_{s_1}(z_1) + \epsilon \boldsymbol{u}^* + C \subseteq \boldsymbol{f}(x) + C .$$

According to Lemma 2, under event $E$, $\boldsymbol{f}(z_2) \in \boldsymbol{R}_{s_2}(z_2)$. Hence, it holds that

$$\boldsymbol{f}(z_2) + \epsilon \boldsymbol{u}^* \in \boldsymbol{f}(x) + C .$$

So, if $z_2 \in \hat{\mathcal{P}}$, then the claim holds with $z = z_2$. If $z_2 \notin \hat{\mathcal{P}}$, then $z_2$ must have been discarded at some round $s_3 \geq s_2$. Because VOGP discards from the set $S_t \setminus \mathcal{P}_{\text{pess},t}$ at $t$, $z_2 \notin \mathcal{P}_{\text{pess},s_3}$. Then, using the definition of the pessimistic set, there exists $z_3 \in \mathcal{A}_{s_3}$ such that

$$\boldsymbol{R}_{s_3}(z_3) \subseteq \boldsymbol{R}_{s_3}(z_2) + C \implies \boldsymbol{R}_{s_3}(z_3) \subseteq \boldsymbol{R}_{s_2}(z_2) + C , \tag{21}$$

where (21) follows from the fact that $\boldsymbol{R}_{s_3}(z_2) \subseteq \boldsymbol{R}_{s_2}(z_2)$. Continuing from (19), we have

$$\boldsymbol{R}_{s_2}(z_2) \subseteq \boldsymbol{R}_{s_2}(z_1) + C \implies \boldsymbol{R}_{s_2}(z_2) + C \subseteq \boldsymbol{R}_{s_2}(z_1) + C + C \tag{22}$$
$$\implies \boldsymbol{R}_{s_2}(z_2) + C \subseteq \boldsymbol{R}_{s_2}(z_1) + C , \tag{23}$$

where (22) follows from the definition of Minkowski sum and (23) follows from the convexity property of $C$.

Combining (21) and (23), we have

$$\boldsymbol{R}_{s_3}(z_3) \subseteq \boldsymbol{R}_{s_2}(z_1) + C . \tag{24}$$

Next, using (24), $s_2 \geq s_1$, and (20), we get

$$\boldsymbol{R}_{s_3}(z_3) + \epsilon \boldsymbol{u}^* \subseteq \boldsymbol{R}_{s_2}(z_1) + \epsilon \boldsymbol{u}^* + C \subseteq \boldsymbol{R}_{s_1}(z_1) + \epsilon \boldsymbol{u}^* + C \subseteq \boldsymbol{f}(x) + C .$$

According to Lemma 2, $\boldsymbol{f}(z_3) \in \boldsymbol{R}_{s_3}(z_3)$ under event $E$. Hence, under event $E$, we have

$$\boldsymbol{f}(z_3) + \epsilon \boldsymbol{u}^* \in \boldsymbol{f}(x) + C .$$

So, if $z_3 \in \hat{\mathcal{P}}$, then the claim holds with $z = z_3$. If $z_3 \notin \hat{\mathcal{P}}$, a similar argument can be made until $z_n \in \hat{\mathcal{P}}$. In the worst case, there comes a point where $z_n \in A_{t_s}$, in which case it is either discarded or added to the $\hat{\mathcal{P}}$. If it is discarded, then it is removed from $\mathcal{P}_{\text{pess},t_s}$ by some design which is then moved to $\hat{\mathcal{P}}$. If it is not discarded, then $z_n \in \hat{\mathcal{P}}$. $\qquad \square$

**Lemma 5.** *Under event $E$, the set $\hat{\mathcal{P}}$ returned by VOGP satisfies condition (ii) in Definition 1.*

*Proof.* We will show that if $\Delta_x^* > 2\epsilon$, then $x \notin \hat{\mathcal{P}} \setminus P^*$. To prove this by contradiction, suppose that $\Delta_x^* > 2\epsilon$ for a design $x \in \hat{\mathcal{P}} \setminus P^*$. By definition of $m(x, x')$, this means that there exists $x' \in P^*$ such that $m(x, x') > 2\epsilon$. Since $x \in \hat{\mathcal{P}}$, it must have been added to $\hat{\mathcal{P}}$ at some round $t$. By the $\epsilon$-covering rule of VOGP, that means for all $z \in \mathcal{W}_t$, $(\boldsymbol{R}_t(x) + \epsilon \boldsymbol{u}^* + C) \cap (\boldsymbol{R}_t(z) - C) = \emptyset$.

We complete the proof by considering two cases:

- **Case 1**: $x' \in \mathcal{W}_t$.
- **Case 2**: $x' \notin \mathcal{W}_t$.

**Case 1**: If $x' \in \mathcal{W}_t$, then

$$(\boldsymbol{R}_t(x) + \epsilon \boldsymbol{u}^* + C) \cap (\boldsymbol{R}_t(x') - C) = \emptyset . \tag{25}$$

By the properties of the Minkowski sum, we have $\boldsymbol{R}_t(x) + \epsilon \boldsymbol{u}^* \subset \boldsymbol{R}_t(x) + \epsilon \boldsymbol{u}^* + C$. This together with (25) results in

$$(\boldsymbol{R}_t(x) + \epsilon \boldsymbol{u}^*) \cap (\boldsymbol{R}_t(x') - C) = \emptyset . \tag{26}$$

According to Lemma 2, under the good event $E$, $\boldsymbol{f}(x) \in \boldsymbol{R}_t(x)$ and $\boldsymbol{f}(x') \in \boldsymbol{R}_t(x')$. Combining this with (26), we conclude that

$$\boldsymbol{f}(x) + \epsilon \boldsymbol{u}^* \notin \boldsymbol{f}(x') - C . \tag{27}$$

Because $\boldsymbol{f}(x) + \epsilon \boldsymbol{u}^* \notin \boldsymbol{f}(x') - \text{int}(C)$, by the definition of $m(x, x')$, $m(x, x') \leq \epsilon$ and we get a contradiction for the case of $x' \in \mathcal{W}_t$.

**Case 2**: If $x' \notin \mathcal{W}_t$, it must have been discarded at an earlier round $s_1 < t$. By the discarding rule, $\exists z_1 \in \mathcal{P}_{\text{pess},s_1}$ such that

$$\boldsymbol{R}_{s_1}(z_1) + \epsilon \boldsymbol{u}^* \subseteq \boldsymbol{y} + C, \ \forall \boldsymbol{y} \in \boldsymbol{R}_{s_1}(x') . \tag{28}$$

We proceed in Case 2, by considering the following two cases based on the status of $z_1$:

- **Case 2.1**: $z_1 \in \mathcal{W}_t$.
- **Case 2.1**: $z_1 \notin \mathcal{W}_t$.

**Case 2.1**: If $z_1 \in \mathcal{W}_t$, since $\boldsymbol{R}_t(z_1) \subseteq \boldsymbol{R}_{s_1}(z_1)$, (28) also implies that

$$\boldsymbol{R}_t(z_1) + \epsilon \boldsymbol{u}^* \subseteq \boldsymbol{y} + C, \ \forall \boldsymbol{y} \in \boldsymbol{R}_{s_1}(x')$$
$$\iff \forall \boldsymbol{y}_{x',s_1} \in \boldsymbol{R}_{s_1}(x'), \ \forall \boldsymbol{y}_{z_1,t} \in \boldsymbol{R}_t(z_1) : \boldsymbol{y}_{z_1,t} \in \boldsymbol{y}_{x',s_1} - \epsilon \boldsymbol{u}^* + C$$
$$\iff \forall \boldsymbol{y}_{x',s_1} \in \boldsymbol{R}_{s_1}(x'), \ \forall \boldsymbol{y}_{z_1,t} \in \boldsymbol{R}_t(z_1) : \boldsymbol{y}_{x',s_1} - \epsilon \boldsymbol{u}^* \preceq_C \boldsymbol{y}_{z_1,t} . \tag{29}$$

Since $z_1 \in \mathcal{W}_t$, the $\epsilon$-covering rule at round $t$ between $x$ and $z_1$ pairs should hold. Combined with the fact that $\boldsymbol{R}_t(x) + \epsilon \boldsymbol{u}^* \subset \boldsymbol{R}_t(x) + \epsilon \boldsymbol{u}^* + C$, it holds that

$$(\boldsymbol{R}_t(x) + \epsilon \boldsymbol{u}^* + C) \cap (\boldsymbol{R}_t(z_1) - C) = \emptyset \tag{30}$$
$$\implies (\boldsymbol{R}_t(x) + \epsilon \boldsymbol{u}^*) \cap (\boldsymbol{R}_t(z_1) - C) = \emptyset$$
$$\iff \forall \boldsymbol{y}_{z_1,t} \in \boldsymbol{R}_t(z_1), \forall \boldsymbol{y}_{x,t} \in \boldsymbol{R}_t(x) : \boldsymbol{y}_{x,t} + \epsilon \boldsymbol{u}^* \npreceq_C \boldsymbol{y}_{z_1,t} . \tag{31}$$

Then, by combining (29) and (31), we get $\boldsymbol{y}_{x,t} + \epsilon \boldsymbol{u}^* \npreceq_C \boldsymbol{y}_{x',s_1} - \epsilon \boldsymbol{u}^*$, $\forall \boldsymbol{y}_{x,t} \in \boldsymbol{R}_t(x)$ and $\forall \boldsymbol{y}_{x',s_1} \in \boldsymbol{R}_{s_1}(x')$. Therefore, according to Lemma 2, under the good event $\mathcal{F}_1$ it holds that $\boldsymbol{f}(x) + 2\epsilon \boldsymbol{u}^* \npreceq_C \boldsymbol{f}(x')$. Since $2\epsilon \boldsymbol{u}^* \in \mathbb{B}(2\epsilon) \cap C$, by the definition of $m(x, x')$, $m(x, x') \leq 2\epsilon$ which is a contradiction.

**Case 2.2**: Next, we examine the case $z_1 \notin \mathcal{W}_t$. Particularly, consider the collection of designs denoted by $z_1, \ldots, z_{n-1}, z_n$ where $z_i$ has been discarded at some round $s_{i+1}$ by being removed from $\mathcal{P}_{\text{pess},s_{i+1}}$ by $z_{i+1}$, as they fulfill the condition $\boldsymbol{R}_{s_{i+1}}(z_{i+1}) \subseteq \boldsymbol{R}_{s_{i+1}}(z_i) + C$. Assume that

$z_n \in \mathcal{W}_t$. Notice that it's always possible to identify such a design, because of the way the sequence was defined. Due to the definition of the pessimistic Pareto set $\mathcal{P}_{\text{pess}}$ given in Definition 3 and (17), we observe that the following set operations hold.

$$\boldsymbol{R}_{s_2}(z_1) + C \supseteq \boldsymbol{R}_{s_2}(z_2) \supseteq \boldsymbol{R}_{s_3}(z_2)$$
$$\boldsymbol{R}_{s_3}(z_2) + C \supseteq \boldsymbol{R}_{s_3}(z_3) \supseteq \boldsymbol{R}_{s_4}(z_3)$$
$$\vdots$$
$$\boldsymbol{R}_{s_{n-1}}(z_{n-2}) + C \supseteq \boldsymbol{R}_{s_{n-1}}(z_{n-1}) \supseteq \boldsymbol{R}_{s_n}(z_{n-1})$$
$$\boldsymbol{R}_{s_n}(z_{n-1}) + C \supseteq \boldsymbol{R}_{s_n}(z_n) \supseteq \boldsymbol{R}_t(z_n) \,. \tag{32}$$

In particular, (32) holds since $t \geq s_n$.

By the definition of Minkowski sum and the convexity of $C$, for any $i \in \{2, \ldots, n-1\}$, we have

$$\boldsymbol{R}_{s_i}(z_{i-1}) + C \supseteq \boldsymbol{R}_{s_{i+1}}(z_i) \implies \boldsymbol{R}_{s_i}(z_{i-1}) + C + C \supseteq \boldsymbol{R}_{s_{i+1}}(z_i) + C$$
$$\implies \boldsymbol{R}_{s_i}(z_{i-1}) + C \supseteq \boldsymbol{R}_{s_{i+1}}(z_i) + C \,.$$

Hence, it holds that

$$\boldsymbol{R}_{s_2}(z_1) + C \supseteq \boldsymbol{R}_{s_3}(z_2) + C$$
$$\boldsymbol{R}_{s_3}(z_2) + C \supseteq \boldsymbol{R}_{s_4}(z_3) + C$$
$$\vdots$$
$$\boldsymbol{R}_{s_{n-1}}(z_{n-2}) + C \supseteq \boldsymbol{R}_{s_n}(z_{n-1}) + C$$
$$\implies \boldsymbol{R}_{s_2}(z_1) + C \supseteq \boldsymbol{R}_{s_n}(z_{n-1}) + C \,. \tag{33}$$

Using the fact that $\boldsymbol{R}_{s_1}(z_1) \supseteq \boldsymbol{R}_{s_2}(z_1)$ when $s_1 \leq s_2$, (33), and (32) in order, we obtain

$$\boldsymbol{R}_{s_1}(z_1) + C \supseteq \boldsymbol{R}_{s_2}(z_1) + C \supseteq \boldsymbol{R}_{s_n}(z_{n-1}) + C \supseteq \boldsymbol{R}_t(z_n) \,. \tag{34}$$

Next, by combining (28) and (34), and using the properties of Minkowski sum, we have

$$\forall \boldsymbol{y} \in \boldsymbol{R}_{s_1}(x') : \boldsymbol{R}_{s_1}(z_1) + \epsilon \boldsymbol{u}^* \subseteq \boldsymbol{y} + C \iff \forall \boldsymbol{y} \in \boldsymbol{R}_{s_1}(x') : \boldsymbol{R}_{s_1}(z_1) \subseteq \boldsymbol{y} - \epsilon \boldsymbol{u}^* + C$$
$$\implies \forall \boldsymbol{y} \in \boldsymbol{R}_{s_1}(x') : \boldsymbol{R}_{s_1}(z_1) + C \subseteq \boldsymbol{y} - \epsilon \boldsymbol{u}^* + C$$
$$\implies \forall \boldsymbol{y} \in \boldsymbol{R}_{s_1}(x') : \boldsymbol{R}_t(z_n) \subseteq \boldsymbol{y} - \epsilon \boldsymbol{u}^* + C \,. \tag{35}$$

Alternatively, (35) can be re-written as

$$\forall \boldsymbol{y}_{x',s_1} \in \boldsymbol{R}_{s_1}(x'), \forall \boldsymbol{y}_{z_n,t} \in \boldsymbol{R}_t(z_n) : \boldsymbol{y}_{x',s_1} - \epsilon \boldsymbol{u}^* \preceq_C \boldsymbol{y}_{z_n,t} \,. \tag{36}$$

Since $z_n \in \mathcal{W}_t$, the $\epsilon$-covering rule at round $t$ between $x$ and $z_n$ pairs should hold. Combined with $\boldsymbol{R}_t(x) + \epsilon \boldsymbol{u}^* \subset \boldsymbol{R}_t(x) + \epsilon \boldsymbol{u}^* + C$, it holds that

$$(\boldsymbol{R}_t(x) + \epsilon \boldsymbol{u}^* + C) \cap (\boldsymbol{R}_t(z_n) - C) = \emptyset \tag{37}$$
$$\implies (\boldsymbol{R}_t(x) + \epsilon \boldsymbol{u}^*) \cap (\boldsymbol{R}_t(z_n) - C) = \emptyset$$
$$\implies \forall \boldsymbol{y}_{z_n,t} \in \boldsymbol{R}_t(z_n), \forall \boldsymbol{y}_{x,t} \in \boldsymbol{R}_t(x) : \boldsymbol{y}_{x,t} + \epsilon \boldsymbol{u}^* \not\preceq_C \boldsymbol{y}_{z_n,t} \,. \tag{38}$$

Then, by combining (36) and (38), we get $\boldsymbol{y}_{x,t} + \epsilon \boldsymbol{u}^* \not\preceq_C \boldsymbol{y}_{x',s_1} - \epsilon \boldsymbol{u}^*$, $\forall \boldsymbol{y}_{x,t} \in \boldsymbol{R}_t(x)$ and $\forall \boldsymbol{y}_{x',s_1} \in \boldsymbol{R}_{s_1}(x')$. Then, according to Lemma 2, under the good event $E$ it holds that $\boldsymbol{f}(x) + 2\epsilon \boldsymbol{u}^* \not\preceq_C \boldsymbol{f}(x')$. Since $2\epsilon \boldsymbol{u}^* \in \mathbb{B}(2\epsilon) \cap C$, by the definition of $m(x, x')$, $m(x, x') \leq 2\epsilon$ which is a contradiction. $\qquad \square$

**Lemma 6.** *If* $\overline{\omega}_t = \max_{x \in \mathcal{W}_t} \omega_t(x) \leq \frac{\epsilon}{2d(1)}$, *then it holds that* $\forall x \in \mathcal{W}_t, \forall \boldsymbol{y}_x, \boldsymbol{y}_{\tilde{x}} \in R_t(x) : \boldsymbol{y}_x \preceq_C \boldsymbol{y}_{\tilde{x}} + \epsilon \boldsymbol{u}^*$.

*Proof.* We have

$$(\boldsymbol{y}_x - \boldsymbol{y}_{\tilde{x}}) \in \mathbb{B}\left(\frac{2\epsilon}{2d(1)}\right) = \mathbb{B}\left(\frac{\epsilon}{d(1)}\right) \,.$$

Therefore, it holds that

$$(\boldsymbol{y}_x - \boldsymbol{y}_{\tilde{x}} + C) \cap \mathbb{S}^{M-1}\left(\frac{\epsilon}{d(1)}\right) \neq \emptyset \iff \exists \tilde{\boldsymbol{\gamma}} \in \mathbb{S}^{M-1}\left(\frac{\epsilon}{d(1)}\right) : \boldsymbol{y}_x - \boldsymbol{y}_{\tilde{x}} \preccurlyeq_C \tilde{\boldsymbol{\gamma}} . \quad (39)$$

By Definition 4, we have

$$\boldsymbol{u}^* = \frac{\underset{z \in A(1)}{\operatorname{argmin}} \left(\|\boldsymbol{z}\|_2\right)}{d(1)} \implies d(1)\boldsymbol{u}^* \in A(1)$$

$$\implies \boldsymbol{u}^* \in \frac{A(1)}{d(1)} \quad (40)$$

$$\implies \boldsymbol{u}^* \in A\left(\frac{1}{d(1)}\right) \quad (41)$$

$$\implies \epsilon\boldsymbol{u}^* \in A\left(\frac{\epsilon}{d(1)}\right), \quad (42)$$

where (40), (41), and (42) follow from Lemma 3. The division in (41) denotes the scaling of the elements of $A(1)$. The proof is completed by observing that, by Definition 4,

$$\forall \boldsymbol{\gamma} \in \mathbb{S}^{M-1}\left(\frac{\epsilon}{d(1)}\right), \forall \boldsymbol{k} \in A\left(\frac{\epsilon}{d(1)}\right) : \boldsymbol{\gamma} \preccurlyeq_C \boldsymbol{k} \quad (43)$$

$$\implies \tilde{\boldsymbol{\gamma}} \preccurlyeq_C \epsilon\boldsymbol{u}^* \quad (44)$$

$$\implies \boldsymbol{y}_x - \boldsymbol{y}_{\tilde{x}} \preccurlyeq_C \epsilon\boldsymbol{u}^* , \quad (45)$$

where (44) follows from $\tilde{\boldsymbol{\gamma}} \in \mathbb{S}^{M-1}\left(\frac{\epsilon}{d(1)}\right)$ and (42), and (45) follows from combining (44) with (39). $\qquad \square$

**Lemma 7.** *VOGP terminates at round $t$ if $\overline{\omega}_t < \frac{\epsilon}{2d(1)}$.*

*Proof.* Let $\mathcal{S}_{t,0}\left(P_{t,0}\right)$, $\mathcal{S}_{t,1}\left(P_{t,1}\right)$ and $\mathcal{S}_{t,2}\left(P_{t,2}\right)$ denote sets $\mathcal{S}_t\left(\mathcal{P}\right)$ at the end of modeling, discarding and covering phases, respectively. For an $x \in \mathcal{S}_{t,0} \setminus P_{t,2}$, if $x \notin \mathcal{S}_{t,1}$, that means it must have been discarded at round $t$. So, if the claim holds, any $x \in \mathcal{S}_{t,0}$ is either discarded or moved to $P_{t,2}$. In order to prove the lemma, we will show that if $x \in \mathcal{S}_{t,0} \setminus P_{t,2}$ holds, then $x$ cannot belong to $S_{t,1}$. To prove this by contradiction, assume that $x \in S_{t,1}$ (otherwise it is discarded). Since $x \in \mathcal{S}_{t,0} \setminus P_{t,2}$, $x$ has not been added to $P_{t,2}$. According to the $\epsilon$-covering rule, there exists some $z^* \in P_{t,1} \cup \mathcal{S}_{t,1}$ for which

$$\exists \boldsymbol{y}_{z^*} \in \boldsymbol{R}_t(z^*), \exists \boldsymbol{y}_x \in \boldsymbol{R}_t(x) : \boldsymbol{y}_x + \epsilon\boldsymbol{u}^* \preccurlyeq_C \boldsymbol{y}_{z^*} \quad (46)$$

Fix $\boldsymbol{y}_{z^*}$ and $\boldsymbol{y}_x$ as given in (46). Since we assume that $x \in P_{t,1} \subseteq P_{t,1} \cup \mathcal{S}_{t,1} = \mathcal{W}_t$, by Lemma 6, we have

$$\forall \boldsymbol{y}_{\tilde{x}} \in \boldsymbol{R}_t(x) : \boldsymbol{y}_{\tilde{x}} \preccurlyeq_C \boldsymbol{y}_x + \epsilon\boldsymbol{u}^* . \quad (47)$$

Again, by Lemma 6, since $z^* \in \mathcal{W}_t$, we have

$$\forall \tilde{\boldsymbol{y}}_z \in \boldsymbol{R}_t(z^*) : \boldsymbol{y}_{z^*} \preccurlyeq_C \tilde{\boldsymbol{y}}_z + \epsilon\boldsymbol{u}^* . \quad (48)$$

Then, starting from (46), and using (48), we have

$$\boldsymbol{y}_x + \epsilon\boldsymbol{u}^* \preccurlyeq_C \boldsymbol{y}_{z^*} \preccurlyeq_C \tilde{\boldsymbol{y}}_z + \epsilon\boldsymbol{u}^*, \ \forall \tilde{\boldsymbol{y}}_z \in \boldsymbol{R}_t(z^*) \quad (49)$$

$$\iff \boldsymbol{y}_x \preccurlyeq_C \tilde{\boldsymbol{y}}_z, \ \forall \tilde{\boldsymbol{y}}_z \in \boldsymbol{R}_t(z^*) . \quad (50)$$

Now, starting from (46) and using (47), we have

$$\boldsymbol{y}_{\tilde{x}} \preccurlyeq_C \boldsymbol{y}_x + \epsilon\boldsymbol{u}^* \preccurlyeq_C \boldsymbol{y}_{z^*}, \ \forall \boldsymbol{y}_{\tilde{x}} \in \boldsymbol{R}_t(x)$$

$$\implies \boldsymbol{y}_{\tilde{x}} \preccurlyeq_C \boldsymbol{y}_{z^*}, \ \forall \boldsymbol{y}_{\tilde{x}} \in \boldsymbol{R}_t(x) . \quad (51)$$

Starting from (51) and using (48), we have

$$\boldsymbol{y}_{\tilde{x}} \preccurlyeq_C \boldsymbol{y}_{z^*} \preccurlyeq_C \tilde{\boldsymbol{y}}_z + \epsilon\boldsymbol{u}^*, \ \forall \boldsymbol{y}_{\tilde{x}} \in \boldsymbol{R}_t(x) \text{ and } \forall \tilde{\boldsymbol{y}}_z \in \boldsymbol{R}_t(z^*)$$

$$\implies \boldsymbol{y}_{\tilde{x}} \preccurlyeq_C \tilde{\boldsymbol{y}}_z + \epsilon\boldsymbol{u}^*, \ \forall \boldsymbol{y}_{\tilde{x}} \in \boldsymbol{R}_t(x) \text{ and } \forall \tilde{\boldsymbol{y}}_z \in \boldsymbol{R}_t(z^*) . \quad (52)$$

Notice that (50) shows that $x \notin \mathcal{P}_{\text{pess},t}$. If $z^* \in \mathcal{P}_{\text{pess},t}$, then (52) shows that $x$ must be discarded. If $z^* \notin \mathcal{P}_{\text{pess},t}$, then $\exists z' \in \mathcal{A}_t$ such that

$$\forall \boldsymbol{y}_{z'} \in \boldsymbol{R}_t(z'), \exists \boldsymbol{y}_{z''} \in \boldsymbol{R}_t(z^*) : \boldsymbol{y}_{z''} \preccurlyeq_C \boldsymbol{y}_{z'} \tag{53}$$

$$\iff \forall \boldsymbol{y}_{z'} \in \boldsymbol{R}_t(z'), \exists \boldsymbol{y}_{z''} \in \boldsymbol{R}_t(z^*) : \boldsymbol{y}_{z''} + \epsilon \boldsymbol{u}^* \preccurlyeq_C \boldsymbol{y}_{z'} + \epsilon \boldsymbol{u}^* . \tag{54}$$

Fix $\boldsymbol{y}_{z''}$ as given in (53). Starting from (52) and using (54), we have

$$\boldsymbol{y}_{\tilde{x}} \preccurlyeq_C \boldsymbol{y}_{z''} + \epsilon \boldsymbol{u}^* \preccurlyeq_C \boldsymbol{y}_{z'} + \epsilon \boldsymbol{u}^*, \forall \boldsymbol{y}_{z'} \in \boldsymbol{R}_t(z') \text{ and } \forall \boldsymbol{y}_{\tilde{x}} \in \boldsymbol{R}_t(x) \tag{55}$$

$$\implies \boldsymbol{y}_{\tilde{x}} \preccurlyeq_C \boldsymbol{y}_{z'} + \epsilon \boldsymbol{u}^*, \forall \boldsymbol{y}_{z'} \in \boldsymbol{R}_t(z') \text{ and } \forall \boldsymbol{y}_{\tilde{x}} \in \boldsymbol{R}_t(x) . \tag{56}$$

(56) shows that if $z' \in \mathcal{P}_{\text{pess},t}$, $x$ should be discarded. If $z' \notin \mathcal{P}_{\text{pess},t}$, a similar argument can be made until the condition to be inside $\mathcal{P}_{\text{pess},t}$ holds since $\mathcal{S}_t$ is a finite set. Hence, the lemma is proved. $\qquad\square$

**Lemma 8.** *Let $t_s$ represent the round in which the algorithm terminates. We have*

$$\sum_{t=1}^{t_s} \overline{\omega}_t \leq \sqrt{t_s \left( 8\beta_{t_s} \sigma^2 \eta M \gamma_{t_s} \right)} ,$$

*where $\eta = \frac{\sigma^{-2}}{\ln(1+\sigma^{-2})}$ and $\gamma_{t_s}$ is the maximum information gain in $t_s$ evaluations.*

*Proof.* Since the diagonal distance of the hyperrectangle $\boldsymbol{Q}_t(x)$ is the largest distance between any two points in the hyperrectangle, we have

$$\sum_{t=1}^{t_s} \overline{\omega}_t^2 = \sum_{t=1}^{t_s} \max_{\boldsymbol{y},\boldsymbol{y}' \in \boldsymbol{R}_t(x_t)} \| \boldsymbol{y} - \boldsymbol{y}' \|_2^2 \tag{57}$$

$$\leq \sum_{t=1}^{t_s} \max_{\boldsymbol{y},\boldsymbol{y}' \in \boldsymbol{R}_{t-1}(x_t)} \| \boldsymbol{y} - \boldsymbol{y}' \|_2^2$$

$$\leq \sum_{t=1}^{t_s} \max_{\boldsymbol{y},\boldsymbol{y}' \in \boldsymbol{Q}_{t-1}(x_t)} \| \boldsymbol{y} - \boldsymbol{y}' \|_2^2$$

$$= \sum_{t=1}^{t_s} \sum_{j=1}^{M} \left( 2\beta_{t-1}^{1/2} \sigma_{t-1}^j(x_t) \right)^2 \tag{58}$$

$$\leq 4\beta_{t_s} \sum_{t=1}^{t_s} \sum_{j=1}^{M} (\sigma_{t-1}^j(x_t))^2 \tag{59}$$

$$= 4\beta_{t_s} \sigma^2 \sum_{t=1}^{t_s} \sum_{j=1}^{M} \sigma^{-2} (\sigma_{t-1}^j(x_t))^2$$

$$\leq 4\beta_{t_s} \sigma^2 \eta \left( \sum_{t=1}^{t_s} \sum_{j=1}^{M} \ln\left( 1 + \sigma^{-2} (\sigma_{t-1}^j(x_t))^2 \right) \right) \tag{60}$$

$$\leq 8\beta_{t_s} \sigma^2 \eta M I(\boldsymbol{y}_{[t_s]}; f_{[t_s]}) \tag{61}$$

$$\leq 8\beta_{t_s} \sigma^2 \eta M \gamma_{t_s}, \tag{62}$$

where $\eta := \sigma^{-2} / \ln \left( 1 + \sigma^{-2} \right)$ and $\sigma$ is the noise standard deviation; (57) is due to the definition of $\bar{\omega}_t$; (58) follows from (16); (59) holds since $\beta_t$ is nondecreasing in $t$; (60) follows from the fact that

$s \leq \eta \ln(1+s)$ for all $0 \leq s \leq \sigma^{-2}$ and that we have

$$
\begin{aligned}
&\sigma^{-2} \left( \sigma_{t-1}^j (x_t) \right)^2 \\
&= \sigma^{-2} \left( k^{jj} (x_t, x_t) - \left( \boldsymbol{k}_{[t-1]} (x_t) \left( \boldsymbol{K}_{[t-1]} + \boldsymbol{\Sigma}_{[t-1]} \right)^{-1} \boldsymbol{k}_{[t-1]} (x_t)^{\mathsf{T}} \right)^{jj} \right) \\
&\leq \sigma^{-2} k^{jj} (x_t, x_t) \\
&\leq \sigma^{-2},
\end{aligned}
\tag{63}
$$

where (63) follows from the fact that $k^{jj}(x, x') \leq 1$ for all $x, x' \in \mathcal{X}$, and (61) follows from (Nika et al., 2021, Proposition 1). Finally, by Cauchy-Schwartz inequality, we have

$$
\sum_{t=1}^{t_s} \overline{\omega}_t \leq \sqrt{t_s \sum_{t=1}^{t_s} \overline{\omega}_t^2} \leq \sqrt{t_s \left( 8 \beta_{t_s} \sigma^2 \eta M \gamma_{t_s} \right)} ,
$$

which completes the proof. □

### E.1.1 PROOF OF THEOREM 1

By definition, $\overline{\omega}_t = \omega_t(x_t) \leq \omega_{t-1}(x_t) \leq \max_{x \in \mathcal{W}_{t-1}} \omega_{t-1}(x) = \overline{\omega}_{t-1}$, where the first inequality is due to $\boldsymbol{R}_t(x) \subseteq \boldsymbol{R}_{t-1}(x)$ for $x \in \mathcal{X}$ and the last inequality is due to the selection rule of VOGP. Hence, we conclude that $\overline{\omega}_t \leq \overline{\omega}_{t-1}$. By above and by Lemma 8, we have

$$
\overline{\omega}_{t_s} \leq \frac{\sum_{t=1}^{t_s} \overline{\omega}_t}{t_s} \leq \sqrt{\frac{8 \beta_{t_s} \sigma^2 \eta M \gamma_{t_s}}{t_s}} .
$$

## E.2 DERIVATION OF THEOREM 2

### E.2.1 AUXILIARY LEMMATA

**Lemma 9.** *(Broxson, 2006, Theorem 15) Let $\boldsymbol{A} \in \mathbb{R}^{n \times n}$, $\boldsymbol{B} \in \mathbb{R}^{m \times m}$ be two real square matrices, where $m, n \in \mathbb{N}$. If $\lambda$ is an eigenvalue of $\boldsymbol{A}$ with corresponding eigenvector $\boldsymbol{v} \in \mathbb{R}^n$ and $\mu$ is an eigenvalue of $\boldsymbol{B}$ with corresponding eigenvector $\boldsymbol{u} \in \mathbb{R}^m$, then $\lambda\mu$ is an eigenvalue of $\boldsymbol{A} \otimes \boldsymbol{B} \in \mathbb{R}^{mn \times mn}$, the Kronecker product of $\boldsymbol{A}$ and $\boldsymbol{B}$, with corresponding eigenvector $\boldsymbol{v} \otimes \boldsymbol{u} \in \mathbb{R}^{mn}$. Moreover, the set of eigenvalues of $\boldsymbol{A} \otimes \boldsymbol{B}$ is $\{\lambda_i \mu_j : i \in [n], j \in [m]\}$, where $\lambda_1, \ldots, \lambda_n$ are the eigenvalues of $\boldsymbol{A}$ and $\mu_1, \ldots, \mu_m$ are the eigenvalues of $\boldsymbol{B}$ (including algebraic multiplicities). In particular, the set of eigenvalues of $\boldsymbol{A} \otimes \boldsymbol{B}$ is the same as the set of eigenvalues of $\boldsymbol{B} \otimes \boldsymbol{A}$.*

**Lemma 10.** *(Chatzigeorgiou, 2013b, Theorem 1) The Lambert function $W_{-1}$ satisfies*

$$
-1 - \sqrt{2u} - u < W_{-1} \left( -e^{-u-1} \right) < -1 - \sqrt{2u} - \frac{2}{3} u
$$

*for every $u > 0$.*

**Lemma 11.** *Let $\tau \geq 0$ and $t > e^\tau$. Setting $\omega \coloneqq \frac{\ln^\tau(t)}{t}$, we have*

$$
t \leq \exp \left( \tau \sqrt{2 \left( \ln(\tau) - \frac{\ln(\omega)}{\tau} - 1 \right)} \right) \frac{\tau^\tau}{\omega} .
\tag{64}
$$

*Proof.* For convenience, let us write $a \coloneqq \ln(t)$. Hence, $\omega = e^{-a} a^\tau$ and

$$
a e^{-\frac{a}{\tau}} = \omega^{\frac{1}{\tau}} .
$$

Let $b \coloneqq \frac{a}{\tau}$. Then,

$$
b \tau e^{-b} = \omega^{\frac{1}{\tau}} ,
$$

which implies that

$$
-b e^{-b} = -\frac{\omega^{\frac{1}{\tau}}}{\tau} .
$$

Thus,

$$-b = W_{-1}\left(-\frac{\omega^{\frac{1}{\tau}}}{\tau}\right).$$

Here, we use the Lambert $W$ function with index $-1$ thanks to the assumption $t > e^\tau$. Then, we obtain

$$\frac{\ln(t)}{\tau} = -W_{-1}\left(-\frac{\omega^{\frac{1}{\tau}}}{\tau}\right)$$

so that

$$t = e^{\left(-\tau \cdot W_{-1}\left(\frac{-\sqrt[\tau]{\omega}}{\tau}\right)\right)}.$$

To be able to use Theorem 10, we set $u$ via

$$\frac{\sqrt[\tau]{\omega}}{\tau} = e^{-u-1} \iff \frac{\ln(\omega)}{\tau} - \ln(\tau) = -u - 1 \iff u = \ln(\tau) - \frac{\ln(\omega)}{\tau} - 1.$$

First, we prove that $u > 0$. To that end, we show that $t \mapsto w(t) := \frac{\ln^\tau(t)}{t}$ is decreasing for $t > e^\tau$. We have

$$\frac{\partial}{\partial t}\left(\frac{\ln^\tau(t)}{t}\right) = \frac{(\tau - \ln(t))\ln^{\tau-1}(t)}{t^2}.$$

Since $t > e^\tau$, we have $\ln^{\tau-1}(t) > 0$ and $t^2 > 0$. Also notice that $\tau - \ln(t) < 0$ for $t > e^z$. Hence, $t \mapsto w(t)$ is decreasing for $t > e^\tau$. Now, if we take $t = e^\tau$, then we have

$$
\begin{aligned}
u &= \ln(\tau) - \frac{\ln(w(e^\tau))}{\tau} - 1 \\
&= \ln(\tau) - \frac{\ln\left(\frac{\ln^\tau(e^\tau)}{e^\tau}\right)}{\tau} - 1 \\
&= \ln(\tau) - \frac{\ln(\ln^\tau(e^\tau)) - \ln(e^\tau)}{\tau} - 1 \\
&= \ln(\tau) - \frac{\ln(\tau^\tau) - \tau}{\tau} - 1 \\
&= \ln(\tau) - \ln(\tau) + 1 - 1 \\
&= 0
\end{aligned}
$$

Since $\omega(t) = \frac{\ln^\tau(t)}{t}$ is decreasing for $t > e^\tau$, and $u = 0$ when $t = e^\tau$, we can conclude that $u > 0$ whenever $t > e^\tau$.

Therefore, we may apply the bound in Theorem 10 and obtain

$$
\begin{aligned}
t &\leq \exp\left((1 + \sqrt{2u} + u)\tau\right) \\
&= \exp\left(\tau + \tau\sqrt{2\left(\ln(\tau) - \frac{\ln(\omega)}{\tau} - 1\right)} + \tau\left(\ln(\tau) - \frac{\ln(\omega)}{\tau} - 1\right)\right) \\
&= \exp\left(\tau\sqrt{2\left(\ln(\tau) - \frac{\ln(\omega)}{\tau} - 1\right)} + \tau\ln(\tau) - \ln(\omega)\right) \\
&= \exp\left(\tau\sqrt{2\left(\ln(\tau) - \frac{\ln(\omega)}{\tau} - 1\right)}\right) \cdot \frac{\tau^\tau}{\omega},
\end{aligned}
$$

which completes the proof. $\qquad\square$

**Lemma 12.** *Let $\boldsymbol{f}$ be a realization from an $M$-output GP with a separable covariance function of the form $(x, x') \mapsto \boldsymbol{k}(x, x') = [\tilde{k}(x, x')k^*(p, q)]_{p,q\in[M]}$, where $\tilde{k}\colon \mathcal{X} \times \mathcal{X} \to \mathbb{R}$ is an RBF or Matérn kernel for the design space and $k^*\colon [M] \times [M] \to \mathbb{R}$ is a kernel for the objective space. For each $p \in [M]$, let $\Psi_p$ be the maximum information gain for a single output GP whose kernel is $(x, x') \mapsto \tilde{k}(x, x')k^*(p, p)$. Then, we have*

$$I(\boldsymbol{y}_{[t]}; \boldsymbol{f}_{[t]}) \leq M \max_{p\in[M]} \Psi_p.$$

*Proof.* Our proof is similar to the proof of (Li et al., 2022, Theorem 2). The difference comes from the structures of the covariance matrices, where the order of Kronecker products are swapped in our case. (Kronecker product of input and output kernels to form the covariance matrix has swapped order.)

Recall that $\boldsymbol{K}_t = (\boldsymbol{k}\,(x_i, x_j))_{i,j \in [t]}$. Hence, we have

$$
\begin{aligned}
I(\boldsymbol{y}_{[t]}; \boldsymbol{f}_{[t]}) &= H(\boldsymbol{y}_{[t]}) - H(\boldsymbol{y}_{[t]} \mid \boldsymbol{f}_{[t]}) \\
&= \frac{1}{2} \ln \left| 2\pi e \left( \boldsymbol{K}_t + \sigma^2 \boldsymbol{I}_{Mt} \right) \right| - \frac{1}{2} \ln \left| 2\pi e \sigma^2 \boldsymbol{I}_{Mt} \right| \\
&= \frac{1}{2} \ln \left( \frac{\left| 2\pi e \left( \boldsymbol{K}_t + \sigma^2 \boldsymbol{I}_{Mt} \right) \right|}{\left| 2\pi e \sigma^2 \boldsymbol{I}_{Mt} \right|} \right) \\
&= \frac{1}{2} \ln \left| \left( \boldsymbol{K}_t + \sigma^2 \boldsymbol{I}_{Mt} \right) \cdot \left( \sigma^2 \boldsymbol{I}_{Mt} \right)^{-1} \right| \\
&= \frac{1}{2} \ln \left| \boldsymbol{I}_{Mt} + \frac{1}{\sigma^2} \boldsymbol{K}_t \right|.
\end{aligned}
$$

By the separable form of $\boldsymbol{k}$, we have

$$
\boldsymbol{K}_t = [\tilde{k}(x_i, x_j)]_{i,j \in [t]} \otimes [k^*(p, q)]_{p,q \in [M]}.
$$

Hence, by Lemma 9 and using the identity $|\boldsymbol{I} + \boldsymbol{A} \otimes \boldsymbol{B}| = |\boldsymbol{I} + \boldsymbol{B} \otimes \boldsymbol{A}|$, we get

$$
I(\boldsymbol{y}_{[t]}; \boldsymbol{f}_{[t]}) = \frac{1}{2} \ln \left| \boldsymbol{I}_{Mt} + \frac{1}{\sigma^2} [k^*(p, q)]_{p,q \in [M]} \otimes [\tilde{k}(x_i, x_j)]_{i,j[t]} \right|. \tag{65}
$$

Notice that

$$
\begin{aligned}
&\boldsymbol{I}_{Mt} + \frac{1}{\sigma^2} [k^*(q, p)]_{p,q \in [M]} \otimes [\tilde{k}(x_i, x_j)]_{i,j \in [t]} \\
&= \begin{bmatrix}
\boldsymbol{I}_t + k^*(1, 1)[\tilde{k}(x_i, x_j)]_{i,j \in [t]} \sigma^{-2}, & \dots, & k^*(1, M)[\tilde{k}(x_i, x_j)]_{i,j \in [t]} \sigma^{-2} \\
\vdots & & \vdots \\
k^*(M, 1)[\tilde{k}(x_i, x_j)]_{i,j \in [t]} \sigma^{-2}, & \dots, & \boldsymbol{I}_t + k^*(M, M)[\tilde{k}(x_i, x_j)]_{i,j \in [t]} \sigma^{-2}
\end{bmatrix}.
\end{aligned}
$$

Since the matrix itself and all of its diagonal blocks are positive definite symmetric real matrices, we can apply Fischer's inequality and obtain

$$
I(\boldsymbol{y}_{[t]}; \boldsymbol{f}_{[t]}) \leq \frac{1}{2} \sum_{p=1}^{M} \ln \left| \boldsymbol{I}_t + k^*(p, p) \sigma^{-2} [\tilde{k}(x_i, x_j)]_{i,j \in [t]} \right|.
$$

This is actually the sum of mutual informations of single output GPs $f_l \sim \mathcal{GP}(\boldsymbol{0}, k^*(l, l)[\tilde{k}(x_i, x_j)]_{i,j \in [t]})$. Notice that a positive constant multiple of an RBF (resp. Matérn) kernel is still an RBF (resp. Matérn) kernel. Since the mutual information is bounded by maximum information gain, we obtain $I(\boldsymbol{y}_{[t]}; \boldsymbol{f}_{[t]}) \leq M \max_{p \in [M]} \Psi_p$, which completes the proof. $\qquad \square$

### E.2.2 Analysis for an RBF kernel

Now, let us assume that we are using an RBF kernel $\tilde{k}$ for the design space. Then, by Lemma 12 and the bounds on maximum information gain established in Srinivas et al. (2012), we have

$$
I(\boldsymbol{y}_{[t]}; \boldsymbol{f}_{[t]}) \leq M \cdot \mathcal{O} \left( \ln^{D+1}(t) \right) \implies \gamma_t \leq M \cdot \mathcal{O} \left( \ln^{D+1}(t) \right).
$$

Notice that in Theorem 1, as $\epsilon$ goes to 0, $T$ goes to infinity. Therefore, we can use the bounds on maximum information gain established in Srinivas et al. (2012).

We have $\beta_t = 2 \ln \left( M\pi^2 |\mathcal{X}| t^2 / (3\delta) \right)$. Let $\alpha_1 = M\pi^2 |\mathcal{X}| / (3\delta), \alpha_2 = 16\sigma^2 \eta M^2$ and $\alpha_3 = \alpha_2 \phi$ where $\phi$ is the multiplicative constant that comes from the $\mathcal{O}$ notation. We continue by finding an

upper bound on the left of the inequality given in Theorem 1. Note that, for $t \geq 3$, we have $\ln(t) \geq 1$ so that

$$
\begin{aligned}
\sqrt{\frac{8\beta_t \sigma^2 \eta M \gamma_t}{t}} &= \sqrt{\frac{\ln(\alpha_1 t^2) \gamma_t \alpha_2}{tM}} \\
&\leq \sqrt{\frac{\ln(\alpha_1 t^2) \alpha_3 \ln^{D+1}(t)}{t}} \\
&= \sqrt{\frac{[\ln(\alpha_1) + \ln(t^2)] \ln^{D+1}(t)}{t} \cdot \alpha_3} \\
&\leq \sqrt{\frac{[\ln(\alpha_1) \ln(t) + \ln(t^2)] \ln^{D+1}(t)}{t} \cdot \alpha_3} \\
&= \sqrt{\frac{\ln^{D+2}(t)}{t} \cdot \alpha_3 (2 + \ln(\alpha_1))}.
\end{aligned}
$$

(66)

Let us define

$$
\begin{aligned}
t^* &:= \min \left\{ t \in \mathbb{N} \mid t \geq 3, \ \sqrt{\frac{\ln^{D+2}(t)}{t} \alpha_3 (2 + \ln(\alpha_1))} \leq \frac{\epsilon}{2d(1)} \right\} \\
&= \min \left\{ t \in \mathbb{N} \mid t \geq 3, \ \frac{\ln^{D+2}(t)}{t} \leq \frac{\epsilon^2}{4d^2(1)(2 + \ln(\alpha_1))\alpha_3} \right\}.
\end{aligned}
$$

Notice that $t^* \geq T$. Moreover, the definition of $t^*$ also guarantees that

$$
\begin{aligned}
t^* - 1 &\leq \min \left\{ t \in \mathbb{R} \mid t \geq 3, \ \frac{\ln^{D+2}(t)}{t} \leq \frac{\epsilon^2}{4d^2(1)(2 + \ln(\alpha_1))\alpha_3} \right\} \\
&\leq \min \left\{ t \in \mathbb{R} \mid t \geq e^{D+2}, \ \frac{\ln^{D+2}(t)}{t} \leq \frac{\epsilon^2}{4d^2(1)(2 + \ln(\alpha_1))\alpha_3} \right\},
\end{aligned}
$$

which implies that

$$
t^* - 1 \leq \tilde{t},
$$

(67)

where $t = \tilde{t}$ is the unique solution of the equation

$$
\frac{\ln^{D+2}(t)}{t} = \frac{\epsilon^2}{4d^2(1)(2 + \ln(\alpha_1))\alpha_3}
$$

in the region $t \in [e^{D+2}, +\infty)$, thanks to the fact that $t \mapsto \frac{\ln^{D+2}(t)}{t}$ is a decreasing function in this region. Since $T - 1 \leq t^* - 1 \leq \tilde{t}$ and we can safely ignore the $-1$ term, we can use Lemma 11 where $\tau = D + 2$ and $\omega = \frac{\epsilon^2}{4d^2(1)(2 + \ln(\alpha_1))\alpha_3}$. Hence, the analysis for RBF kernel is complete.

### E.2.3 ANALYSIS FOR A MATÉRN KERNEL

Next, we suppose that $\tilde{k}$ is a Matérn kernel. The analysis is very similar to the analysis for an RBF kernel. We use the bounds found by Vakili et al. (2021b):

$$
I(\boldsymbol{y}_{[t]}, \boldsymbol{f}_{[t]}) \leq M \cdot \mathcal{O}\left(T^{\frac{D}{2\nu+D}} \ln^{\frac{2\nu}{2\nu+D}}(T)\right) \implies \gamma_t \leq M \cdot \mathcal{O}\left(T^{\frac{D}{2\nu+D}} \ln^{\frac{2\nu}{2\nu+D}}(T)\right).
$$

By observing only the exponent of the denominator $t$ changes and $D + 1$ is replaced by $\frac{2\nu}{2\nu+d}$ in comparison to (66), we have (67) with $t = \tilde{t}$ solving the equation

$$
\frac{\ln^{4\nu+D}(t)}{t^{(2\nu)/(2\nu+D)}} = \frac{\epsilon^2}{4d^2(1)(2 + \ln(\alpha_1))\alpha_3} \Leftrightarrow \frac{\ln^{\frac{(4\nu+D)(2\nu+D)}{2\nu}}(t)}{t} = \frac{\epsilon^{\frac{2\nu+D}{\nu}}}{(4\alpha_3 d^2(1)(2 + \ln(\alpha_1)))^{\frac{2\nu+D}{2\nu}}}
$$

(68)

The sample complexity is then found by applying Lemma 11 with $\tau = \frac{(4\nu+D)(2\nu+D)}{2\nu}$ and $\omega$ being the last term in (68). Hence, the analysis for a Matérn kernel is complete.

