# OpenReview forum: "Bayesian Vector Optimization with Gaussian Processes"
_ICLR.cc/2024/Conference — Submitted to ICLR 2024_

### Official Review · Reviewer_bF5L · 2023-10-26

**Soundness:** 2 fair
**Presentation:** 2 fair
**Contribution:** 2 fair
**Rating:** 5
**Confidence:** 5

**Summary:**

This paper proposes to use Gaussian processes (GPs) to identify the optimal set under a given preference cone, reducing the sample complexity of such approaches. It relies on the uncertainty quantification features of GPs to filter points that are likely to be dominated as well as optimal ones. The query is at the least certain design, so a fully exploratory scheme. A theoretical analysis is provided as well as some empirical results on data sets.

**Strengths:**

- The use of preference cones is less common in the multi-objective Bayesian optimization community.
- Existing theoretical results are extended to ordering cones.

**Weaknesses:**

- There are a lot of existing works on preference learning with BO.
- It is not clear how to use cones for a practitioner.
- Only discrete input spaces are considered.
- The links to a similar method, PAL, are not detailed enough. An empirical comparison with this method is needed.
- The empirical results are not reproducible.

**Questions:**

Related references:
- Picheny, V. (2015). Multiobjective optimization using Gaussian process emulators via stepwise uncertainty reduction. Statistics and Computing, 25(6), 1265-1280.
- Emmerich, M. T., Deutz, A. H., & Klinkenberg, J. W. (2011, June). Hypervolume-based expected improvement: Monotonicity properties and exact computation. In 2011 IEEE Congress of Evolutionary Computation (CEC) (pp. 2147-2154). IEEE.
- Yang, K., Li, L., Deutz, A., Back, T., & Emmerich, M. (2016, August). Preference-based multiobjective optimization using truncated expected hypervolume improvement. In 2016 12th International Conference on Natural Computation, Fuzzy Systems and Knowledge Discovery (ICNC-FSKD) (pp. 276-281). IEEE.
- Lepird, J. R., Owen, M. P., & Kochenderfer, M. J. (2015). Bayesian preference elicitation for multiobjective engineering design optimization. Journal of Aerospace Information Systems, 12(10), 634-645.
- Khan, F. A., Dietrich, J. P., & Wirth, C. (2022). Efficient Utility Function Learning for Multi-Objective Parameter Optimization with Prior Knowledge. arXiv preprint arXiv:2208.10300.
- Garnett, R. (2023). Bayesian optimization. Cambridge University Press.
- Svenson, J., & Santner, T. (2016). Multiobjective optimization of expensive-to-evaluate deterministic computer simulator models. Computational Statistics & Data Analysis, 94, 250-264.
- Ignatenko, T., Kondrashov, K., Cox, M., & de Vries, B. (2021). On Preference Learning Based on Sequential Bayesian Optimization with Pairwise Comparison. arXiv preprint arXiv:2103.13192.
- Ungredda, J., & Branke, J. (2023, July). When to Elicit Preferences in Multi-Objective Bayesian Optimization. In Proceedings of the Companion Conference on Genetic and Evolutionary Computation (pp. 1997-2003).
- Taylor, K., Ha, H., Li, M., Chan, J., & Li, X. (2021, June). Bayesian preference learning for interactive multi-objective optimisation. In Proceedings of the Genetic and Evolutionary Computation Conference (pp. 466-475).
- Jussi Hakanen and Joshua D Knowles. On using decision maker preferences with ParEGO.
In International Conference on Evolutionary Multi-Criterion Optimization, pages 282–297.
Springer, 2017.
- Barracosa, B., Bect, J., Baraffe, H. D., Morin, J., Fournel, J., & Vazquez, E. (2022). Bayesian multi-objective optimization for stochastic simulators: an extension of the Pareto Active Learning method. arXiv preprint arXiv:2207.03842.

Page 1: It is unclear what the inclusion relation is between Pareto optimal solution and cone order optimal solution (“However, this approach can be restrictive, as it only permits a certain set of trade-offs between objectives.” and then “preference cones provide a way to bias the search toward certain regions of the Pareto front.”

Introductive agricultural example: perhaps you could complement Figure 1 with an actual Pareto front to better illustrate the interest. You could add the pessimistic Pareto front defined later. Also in Figure 1, cones are parameterized with angles but later on with a matrix.

Can you describe the convex optimization problem used in the pessimistic Pareto front construction?

Discuss the relation with epsilon-PAL. It should be added to the empirical comparison.

It is unclear where the cone properties appear in the theoretical results.

Could you provide timings? Progress curves rather than just fixed snapshots? Random search should be added as a baseline.

As I understand, the noise hyperparameter is not learned by the GP? What is multi-output covariance kernel used? Is it the same across all compared methods? Is the learning strategy shared among all methods (e.g., fixed hyperparameters)?
State of the art EHVI is proposed by Daulton, S., Balandat, M., & Bakshy, E. (2020). Differentiable expected hypervolume improvement for parallel multi-objective Bayesian optimization. Advances in Neural Information Processing Systems, 33, 9851-9864.
Could you add an example with more than 2 objectives? Too many details are missing to reproduce the experiments: code used, number of samples, etc.

---

> ### Author Response · Authors · 2023-11-23
>
> We thank the reviewer for their thoughtful comments and valuable insights.
>
> **Weaknesses**
>
> **(... a lot of existing works ...)** In the revised version of the paper, we update the related works part of the introduction by including the existing works suggested by the reviewer.
>
> **(... how to use cones for a practitioner...)**  In [1], authors discuss the use of cones in practice with preferences of decision maker in the form of relative importance (equivalently trade-off constraint). They give an example similar to the farming example in our paper's introduction. In [2], this approach is generalized to arbitrary dimensions, and authors consider the case where a number of criteria are considered to be less important than others. They algebraically characterize the polyhedral ordering cones that correspond to preferences in the mentioned form. In their case, the decision-maker is required to pass on the trade-off constraint among objectives , which is practical.
>
> [1] Hunt, B.J., Wiecek, M.M. (2003). Cones to Aid Decision Making in Multicriteria Programming. In: Multi-Objective Programming and Goal Programming. Advances in Soft Computing, vol 21. Springer, Berlin, Heidelberg. https://doi.org/10.1007/978-3-540-36510-5\_20
>
> [2] Hunt, Brian J. et al. “Relative importance of criteria in multiobjective programming: A cone-based approach.” Eur. J. Oper. Res. 207 (2010): 936-945.
>
> **(The empirical results are not reproducible.)** We note that the code of VOGP and its experiments was submitted through the supplementary material submission.
>
> **(The links to a similar method, PAL ...)** In order to make the connection clearer, we added additional texts in the introduction. As it's mentioned there, when VOGP is considered with the componentwise order, it can be seen as a variant of PAL that can handle dependent objectives. Therefore, we suspected that a comparison with PAL in componentwise order would boil down to dependent versus independent modeling which would not be informative.
>
> **Questions**
>
> **( ... It is unclear what the inclusion relation ...)** We say that a vector weakly dominates another with respect to $C$ if their difference lies in $C$. Since the 90 degree cone is a subset of the 135 degree cone, if the difference of two vectors lies in the 90 degree cone, then it will also lie in the 135 degree cone. Since the Pareto set consists of all non-dominated design , a design that is not dominated under the 135 degree cone will not be dominated under the 90 degree cone. Hence, the Pareto set with respect to the 135 degree cone is a subset of the Pareto set with respect to the 90 degree cone.
>
> **(Can you describe ...)** To see whether a point1 is in pessimistic pareto set, we check if all its vertices of another hyperrectangle (of point2) can be represented by summation of a help vector from the cone and a point from the  points1's hyperrectangle. This is to check the pessimistic Pareto definition. If this holds for any such point2, point1 is not in the pessimistic Pareto set. CVXPY library of python is used to solve the optimization problem.
>
> **(... where the cone properties appear ...)** In both bounds found in both Theorem 1 and Theorem 2, $d(1)$ is a cone-dependent constant that reflects the effects of different ordering cones.
>
> **(... the noise hyperparameter ... )** In the problem definition, a fixed noise variance is assumed. In the experiments, the variance is assumed to be known and is not trained.
>
> **(What is multi-output covariance kernel used?)** The model used in the experiments is the Mulitask model proposed in [3]. The other methods mentioned in the paper were proposed and implemented with independent GPs. Therefore, the GP used for other methods is independent GP, where the relation among objectives are not modelled.
>
>  [3] Bonilla, Edwin V., Kian Chai, and Christopher Williams. "Multi-task Gaussian process prediction." Advances in neural information processing systems 20 (2007).
>
>
> **(Could you add an example with more than 2 objectives?)** We added a real-world dataset to the experiments section that has 6 design dimensions, 4 objective dimensions and 1000 datapoints. The performance results of VOGP on this dataset can be seen in Table 2 of the updated paper.
>
> **(Too many details are missing to reproduce the
> experiments: code used, number of samples, etc.)**
> We note that the code to reproduce the results were shared through supplementary material submission. The number of samples are given in the experiments section under the name ``SC" (e.g., in Table 2).
>
> **(Random search should be added as a baseline.)** We kindly refer the reviewer to Section C of the updated paper.
>
> **(Progress curves rather than just fixed snapshots?)** In the updated paper, we provide progress curves. We kindly refer the reviewer to Figure 3 of the updated paper.
>
> **(State of the art EHVI is proposed by ... )** Fixed.
>
> **(Is the learning strategy shared ...)** Yes.

---

### Official Review · Reviewer_ahkf · 2023-10-29

**Soundness:** 2 fair
**Presentation:** 2 fair
**Contribution:** 3 good
**Rating:** 5
**Confidence:** 2

**Summary:**

Vector optimization formulates a multi-objective optimization problem in a way that expresses a user's preference to trade off the different objectives.
The study leverages the machinery of Bayesian optimization (BayesOpt) for the task of vector optimization, where the objective functions are assumed to be black boxes.
The authors propose a sample-efficient policy that explores the solution space using as few queries as possible.
Using the smoothness assumption made by the Gaussian process (GP), the paper also proves a PAC-learning guarantee for the proposed algorithm.
The experiments show that this algorithm is more effective at maximizing the success rate (according to the PAC criterion) than a wide range of baselines, while keeping the number of queries low.

**Strengths:**

The problem studied in this paper is motivated well.
Vector optimization seems like an elegant way for a user to flexibly express their preference over multiple objectives.
Using BayesOpt to tackle this problem when objectives are expensive to query seems like a natural solution.
The proposed algorithm is proven to have good theoretical guarantee, where the algorithm will return an approximate Pareto-optimal (in the context of vector optimization) with high probability in at most some specified number of queries.
The experiments are convincing in showing that the proposed method is competitive against many baselines in the multi-objective optimization literature.

**Weaknesses:**

I find the background and problem definition a bit hard to follow.
The authors can consider prioritizing intuitive understanding over exposition of all the math.
The same goes for the algorithm itself; perhaps add a diagram on the procedure the policy goes through in Algorithm 1.

The paper does a good job comparing the proposed algorithm against state-of-the-art multi-objective BayesOpt policies.
However, from what I understand, only one algorithm from the vector optimization literature, Naïve Elimination, is included.
It could be worth including other (possibly sample-inefficient) algorithms for a more complete comparison.

The experiments are set up in a way that the GP always has access to the correct hyperparameters (obtained via training on the entire data set).
In many real-life settings, we don't have access to the correct hyperparameters, or even good priors for them.
The paper could benefit from studying the effects of the GP having the wrong hyperparameters on the performance of the algorithm.

**Questions:**

- In Definition 1, my understanding is that the second condition (ii) specifies that $x \in P$ is not dominated by another by more than $2 \epsilon$.
What does the first condition (i) say?
Perhaps the the background of vector optimization could benefit from more descriptive discussions.
- The authors noted that once a point is added to the Pareto set, it will not be removed.
This doesn't match my intuition well; isn't it possible that as we learn more about the objectives, we realize that some of the points already added aren't non-dominated?
- Could the authors comment on the possible difficulties of extending the proposed algorithm to continuous setting?
I imagine the challenge lies in the discarding and Pareto identification phases.
Can we try to discard and identify dominated and non-dominated regions, respectively?

---

> ### Author Response · Authors · 2023-11-23
>
> We thank the reviewer for their thoughtful comments and valuable insights.
>
> **Weaknesses**
>
> **(... prioritizing intuitive
> understanding ...)** In the updated paper, we include a remark (Remark 1) that clarifies what the success conditions mean intuitively. We also provide a diagram that visualizes the process and the steps of Algorithm 1 (Figure 3). We hope these additions will help improve the clarity of the paper.
>
> **(... only one algorithm ...)** To the best of our knowledge, Naive Elimination in Ararat and Tekin (2023) is the only other algorithm that can handle vector optimization problems with polyhedral ordering cones (other than the positive orthant) in the stochastic K-armed bandit framework with observation noise.
>
> In the literature on deterministic vector optimization, there are algorithms that can solve constrained linear and convex vector optimization problems under general polyhedral ordering cones. Below is an incomplete list of works where such algorithms were proposed:
>
> - Löhne, Rudloff, Ulus. “Primal and dual approximation algorithms for convex vector optimization problems.” Journal of Global Optimization 60 (2014): 713-736.
>
>
> - Hamel, Löhne, Rudloff. “Benson type algorithms for linear vector optimization and applications.” Journal of Global Optimization 59 (2014): 811-836.
>
> - Ararat, Ulus, Umer. “A norm minimization-based convex vector optimization algorithm.” Journal of Optimization Theory and Applications 194 (2022): 681-712.
>
> However, none of these algorithms are applicable for our problem as these algorithms assume a completely deterministic and fully observable objective functions that satisfy linearity/convexity assumptions. That is why we could compare our method only with Naive Elimination.
>
> In addition, in the supplementary document, we compare VOGP with the other MOO algorithms in the stochastic bandit framework by post-processing their outputs (that are computed under the usual positive orthant cone) with respect to the cone $C=C_{135^o}$.
>
>
>
> **(... GP always has access to the correct hyperparameters ...)** In the updated supplemental document, we share an ablation study where the empirical effects of unknown or partially known kernel hyperparameters are discussed. We kindly refer the reviewer to Section C.
>
> **Questions**
>
> **(... What does the first condition (i) say? ...)** It is correct that (ii) specifies that  $x \in P$  is not dominated by another by more that $2\epsilon$. (i) specifies that for all $x^* \in P^*$, there exists a $x \in P$ such that $x$ dominates $x^*$ with a help of $\epsilon$ normed vector. So, (i) can be thought as the covering condition.
>
>
> **(... Pareto set, it will not be removed ...)** Pareto Identification and discarding subroutines check the ordering relation for all the possibilities inside confidence regions (i.e., hyperrectangles). By doing so, given the hyperrectangles include the true objective values, VOGP algorithm's Pareto identification subroutine makes sure the added designs are in the Pareto set, no matter how early in the algorithm. This is one of the strengths of VOGP, as designs are labeled as Pareto without waiting for the whole algorithm to terminate. This opens up applications in pipelines where the early identified Pareto designs are moved through the pipeline immediately, increasing the efficiency. Another application that arises from this is the task of identifying only one Pareto design. While other methods might need the whole algorithm to terminate and pick one from the returned Pareto set, VOGP would most efficiently identify the first Pareto design.
>
>
> **(... extending the proposed algorithm to continuous setting ...)**  Yes, the main challenge is that the Pareto Identification and Discarding steps are one to one comparison based steps which does not work in continuous problems where infinitely many comparisons are required. An extension where regions instead of points are considered can indeed be the solution. A practically easy instance of this kind of solution is the method of adaptive discretization where the search space is intelligently divided with points that are representatives of regions around them [1].
>
> [1] Nika, A., Bozgan, K., Elahi, S., Ararat, Ç., \& Tekin, C. (2020). Pareto active learning with Gaussian processes and adaptive discretization. arXiv preprint arXiv:2006.14061.

---

### Official Review · Reviewer_XrYo · 2023-10-31

**Soundness:** 2 fair
**Presentation:** 3 good
**Contribution:** 2 fair
**Rating:** 5
**Confidence:** 3

**Summary:**

This paper proposes a generalization of multi-objective Bayesian optimization.
Specifically, the partial ordering is defined by a convex cone.
The partial ordering induced by coordinate-wise comparisons used in the literature of multi-objective Bayesian optimization is a special case where the convex cone is the nonnegative orthant.
Next, the paper proposes an $(\epsilon, \delta)$-probably approximately  correct algorithm, which finds an approximate Pareto set with high probability.
Theoretically, the authors present an upper bound on the number of iterations finding an $(\epsilon, \delta)$-PAC Pareto set.
Empirical evaluations on a few low dimensional functions demonstrates its superior sample complexity compared to the naive elimination (Ararat and Tekin, 2023), a recent algorithm proposed in the setting of stochastic bandits.

**Strengths:**

- This paper introduces to the BO community the concept of partial ordering induced by a convex cone, which I think is a useful generalization and may be beneficial to certain BO applications.
- A theoretical analysis on the sample complexity is presented, which shows that the algorithm finds an $\epsilon$-approximate Pareto set with high probability $1 - \delta$.

**Weaknesses:**

- The experiments are done on very small datasets. The largest dimension is $4$ and all tasks have two objectives.
- At this point, the method in the paper is restricted to discrete domains $\mathcal{X}$, which limits the application of the method: the constant $\beta_t$ in the algorithm depends on the cardinality of $\mathcal{X}$, the theorem statements assume finite domains, and the evaluation metrics needs a finite cardinality $|\mathcal{X}|$ as well. I would assume this is fixable by extending the results in the paper, but additional empirical evaluations are required.
- The naive elimination is a theoretical construct in the bandit setting, which does not exploit the correlation in the GP model. A more meaningful baseline for comparison is other multi-objective BO algorithms. For example, as the angle $\theta$ changes, how do the Pareto precision and Pareto recall change comparing with regular multi-objective BO algorithms (which are designed specifically for a particular definition of Pareto optimality)? Another helpful experiment is to plot metrics w.r.t. the number of queries. This allows us to visually check the convergence. From the current tables, it is hard to tell if they are fully converged or not.
- The experimental setup is non-standard. For example, "for each dataset, we learn the kernel hyperparameters by training on the entire dataset". However, the hyperparameters in BO are typically learned as more queries are added to the training data. Have the author tried the latter more commonly used setting?

Minor comments:
- The following notations need more explicit definitions in the main text: the hyperrectangles $R_t(x)$, their diameters $\omega_t(x)$, the information gain $\gamma_t$ and the constant $\eta$.
- In the definition of the polyhedral cone, it should be $C = \\{\mathbf{x} \in \mathbb{R}^M: W \mathbf{x} \geq 0\\}$ and $W$ should be $N \times M$. The cone is defined in the output space, not in the domain.

**Questions:**

- Theorem 1 and Theorem 2 need to add an extra technical assumption on the cone $C$. Otherwise the bounds may be vacuous in certain cases. For example, $C = \\{(x_1, x_2) \in \mathbb{R}^2: x_1 = x_2\\}$ is a well-defined polyhedral ordering cone. However $d(1) = \infty$ in this case and thus both bounds become vacuous.
- Can you share more intuition on Definition 4? My intuition is that $\mathbf{u}^*$ points to the "center" of the cone.
- In line 3 of Algorithm 4, why $+ C$ and $- C$ are on both sides of the intersection? Shouldn't $(\mathbf{R}_t(\mathbf{x}) + \epsilon \mathbf{u}^* + C) \cap \mathbf{R}_t(\mathbf{x}^\prime)$ be already sufficient?
- The evaluation metrics PA, PR and PP need the ground truth Pareto set $P^*$. How are the ground truth Pareto sets computed?
- Branin and Currin are continuous functions defined on continuous domains. Does the experiment in Table 1 discretize their domains?

---

> ### Author Response · Authors · 2023-11-23
>
> We thank the reviewer for their thoughtful comments and valuable insights.
>
>
> **Weaknesses**
>
> **(The experiments are done on very small datasets ...)** We added a real-world dataset (MAR dataset) to the experiments section that has 6 design dimensions, 4 objective dimensions and 1000 datapoints. This dataset has the objective of optimizing bulk carrier architecture specifically for navigating the Panama Canal. The performance results of VOGP on this dataset can be seen in Table 2 of the updated paper.
>
>
> **(.... the method in the paper is restricted to discrete domains ...)** Indeed the extension of VOGP to continuous spaces is possible through a method called adaptive discretization like in [1] where the design space is intelligently discretized so that the continuous space is explored well. Though its implementation is simple, carrying theoretical guarantees over to the adaptive discretization version of VOGP requires a more spohisticated theoretical analysis. We view this extension of our work as a promising future research direction.
>
> [1] Nika, A., Bozgan, K., Elahi, S., Ararat, Ç., \& Tekin, C. (2020). Pareto active learning with Gaussian processes and adaptive discretization. arXiv preprint arXiv:2006.14061.
>
>
>
> **(... a more meaningful baseline for comparison  ...)** In the updated supplemental document, we included the comparison of VOGP with other MOO methods under 135 degree cone order. The results of this experiment shows the necessity of cone-dependent algorithm (VOGP) when the used cones are not positive orthant cones (the $90^\circ$ cone). We kindly refer the reviewer to Section A.2.
>
> **(Another helpful experiment is to plot metrics w.r.t. the number of queries. This allows us to visually check the convergence)** In the updated paper, we provide progress curves. We kindly refer the reviewer to Figure 3 of the updated paper.
>
>
>
>
> **(The experimental setup is non-standard ...)** In the updated supplemental document, we investigate the effects of the assumptions on kernel hyperparameters in detail. We consider the cases where kernel hyperparameters are unknown, known and partially known. We share the performance results of VOGP under these cases and discuss the effects. We kindly refer the reviewer to Section C.
>
>
>
>
>
> **Minor Comments**
> **(The following notations ...)** The equation that defines the hyperrectangles is now written more explicitly. The explicit definition of hyperrectangle diagonal is now provided in the explanation of the evaluating phase. The definition of $\eta$ is now provided in Theorem 1. We note that an already explicit definition of $\gamma_t$ was given in Definition 2. We would appreciate if the reviewer could provide details to be added to make it more explicit.
>
>
> **(In the definition of  ...)** Fixed.
>
> **Questions**
>
> **(Theorem 1 and Theorem 2 ...)** By the definition of the ordering cone (proper cone), an ordering cone has non-empty interior (Section 2.4 of [2]). Therefore, the mentioned cone is not an ordering cone that is allowed in our setting.
>
> [2] S. Boyd and L. Vandenberghe, Convex Optimization. Cambridge: Cambridge University Press, 2004.
>
>
> **(... more intuition on Definition 4 ...)** Indeed, $\boldsymbol{u^*}$ points to the ``center" of the cone. More specifically, it points to the center of the sphere or a circle that is tangent to every halfspace that defines $C$. In two dimensions, this corresponds to the center of the incircle (though there are only two lines that define the cone in 2D: given the circle radius, the third edge of the triangle is implicitly specified).
>
>
> **(In line 3 of Algorithm 4 ...)** Fixed.
>
>
>
> **(The evaluation metrics PA, PR and PP ...)** Since the noiseless objective values of designs are known during the calculation of evaluation metrics, we can iterate through designs and check if there are any other designs that dominate them in the dataset. The designs that are not dominated by any other design are the elements of the true Pareto set $P^*$.
>
>
> **(Branin and Currin are continuous functions ...)** Yes, the datapoints were randomly sampled from the continuous functions.

---

### Official Review · Reviewer_vz76 · 2023-10-31

**Soundness:** 3 good
**Presentation:** 3 good
**Contribution:** 3 good
**Rating:** 6
**Confidence:** 2

**Summary:**

This work proposes vector optimization for multiple-objective optimization using confidence intervals built from Gaussian Processes. The proposed method called VOGP allows users to convey objective preferences through ordering cones while performing efficient sampling by exploiting the smoothness of the objective function, resulting in a more effective optimization process that requires fewer evaluations. Both theoretical guarantee and experimental results are demonstrated to consolidate the claims.

**Strengths:**

proposes a vector optimization based on Gaussian processes.

**Weaknesses:**

There might be more applications illustrated.

**Questions:**

N/A

---

> ### Author Response · Authors · 2023-11-23
>
> We thank the reviewer for their thoughtful comments and valuable insights.
>
> **Weaknesses**
>
> (There might be more applications ...) In the experiments section, we added the application of VOGP on another real-world dataset that has six design dimensions and four objective dimensions. This dataset has the objective of optimizing bulk carrier architecture specifically for navigating the Panama Canal. The performance results of VOGP on this dataset can be seen in Table 2 of the updated paper.
>
> For more possible applications of VOGP, we kindly refer the reviewer to our response to the second question of reviewer ahkf.

---

### Author Response · Authors · 2023-11-23

Dear Reviewers,

Thank you for your valuable feedback regarding our initial submission. Here, we summarize minor changes in the paper and their reasons.


We have observed that SR (Success Ratio) was uninformative about returned Pareto set quality due to its strictness. As an example, even if one single returned design does not satisfy the second success condition, the SR score of that whole set becomes 0. To make it more informative, SR is now replaced with SR1 and SR2 which are the frequency of designs that obey the success conditions 1 and 2, respectively. (Formal definitions of SR1 and SR2 can be found in Definition 6 of the updated paper.)

We have relocated the section on evolutionary algorithms in the related works to the supplemental document (Section D). This adjustment allows us to accommodate important updates and maintain the conciseness of the main manuscript.

---

### Meta-Review · Area_Chair_Qko6 · 2023-12-23

**Metareview:**

This work considers multi-objective optimization with partial ordering information via ordering cones.  Reviewers largely gave this paper a weak reject rating. There were a few recurring themes that came up: reviewers felt that the exposition and motivation were too abstract and not of clear practical use. Some reviewers wanted to see more baselines, and the authors provided additional simulation results in the appendix with a variety of MOO baselines during the response period. Ultimately, reviewers felt that many of their critiques were not resolved by the authors' response.

A particularly important area called out by nearly all reviewers is that there are few practical settings in which HPs are known, especially in sample-efficient optimization problems. Performance with unknown HPs was not examined empirically in the initial submission, and the new results in Section C, indicate that the algorithm does not perform particularly well in this case. I would recommend that the authors consider unknown HPs as the main case for their work, and come up with a scheme that performs better when HPs must be estimated online.  Unknown HPs also came up in critiques about the theory. It is fairly standard for theory results in the GP Bandit / Bayesian optimization literature to assume some discrete set of points and known HPs, but many reviewers wanted to see theory for continuous spaces or unknown HPs. If it is possible to say more about these cases theoretically, it would strengthen the paper.

I thank the authors and reviewers for their engagement during this review process and hope that this feedback will in turn improve this line of work.

**Justification For Why Not Higher Score:**

Without covering the case of unknown HPs, this work is not very useful, and this point was raised by nearly all reviewers.  There is also clear room for improvement in the exposition and theory.

**Justification For Why Not Lower Score:**

N/A

---

### Decision · Program_Chairs · 2024-01-16

Reject